# Giant antidamping orbital torque originating from the orbital Rashba-Edelstein effect in ferromagnetic heterostructures

Xi Chen [1], Yang Liu[2], Guang Yang [3], Hui Shi[3], Chen Hu[4], Minghua Li[3] & Haibo Zeng [1]

Enhancing the in-plane current-induced torque efficiency in inversion-symmetry-breaking ferromagnetic heterostructures is of both fundamental and practical interests for emerging magnetic memory device applications. Here, we present an interface-originated magneto-electric effect, the orbital Rashba–Edelstein effect, for realizing large torque efficiency in $Pt/Co/SiO_2/Pt$ films with strong perpendicular magnetic anisotropy (PMA). The key element is a pronounced Co $3d$ orbital splitting due to asymmetric orbital hybridization at the Pt/Co and $Co/SiO_2$ interfaces, which not only stabilizes the PMA but also produces a large orbital torque upon the Co magnetization with current injection. The torque efficiency is found to be strongly magnetization direction- and temperature-dependent, and can reach up to 2.83 at room temperature, which is several times to one order of magnitude larger than those previously reported. This work highlights the active role of the orbital anisotropy for efficient torque generation and indicates a route for torque efficiency optimization through orbital engineering.

[1] MIIT Key Laboratory of Advanced Display Materials and Devices, Institute of Optoelectronics & Nanomaterials, College of Materials Science and Engineering, Nanjing University of Science and Technology, Nanjing 210094, China. [2] Nanoscale Physics & Devices Laboratory, Institute of Physics, Chinese Academy of Sciences, Beijing 100190, China. [3] Department of Materials Physics and Chemistry, University of Science and Technology Beijing, Beijing 100083, China. [4] Center for the Physics of Materials and Department of Physics, McGill University, Montreal, QC H3A 2T8, Canada. Correspondence and requests for materials should be addressed to H.Z. (email: zeng.haibo@njust.edu.cn)

In heavy-metal (HM)/ferromagnet (FM)/oxide heterostructures, which are the core elements of modern magnetic memory technologies and logic devices, rich magnetic and electrical effects can emerge from the coupling between the spin and orbital of electrons. A typical effect is that of the perpendicular magnetic anisotropy (PMA) arising from the combination of orbital hybridization and spin–orbit coupling (SOC) at the HM/FM interface[1] or the FM/oxide interface[2]. Another relevant effect is the SOC-mediated conversion of an in-plane electric current into a spin polarization through the spin Hall effect (SHE) of the HM bulk or the spin Rashba effect (SRE) at the symmetry-breaking interfaces[3–6]. At present, research on the control of magnetization with PMA using the torque from the generated spin polarizations constitute an active field known as spin-orbitronics[7–18], which aims at constructing advanced magnetic devices with high integration density, high read/write speed, and low energy cost.

Despite its technological appeal, manipulating magnetization using the current-induced torque in HM/FM/oxide heterostructures has several challenging issues that still remain to be overcome. The first one is the conflict between high PMA and low critical current density ($j_c$) for current-induced magnetization switching (CIMS) simultaneously. Strong PMA is of great importance for magnetic bit stability and write-in error rate reduction[19] in miniaturized magnetic devices; however, it also requires high $j_c$ for CIMS, because $j_c$ is proportional to effective magnetocrystalline anisotropy (MCA) energy $K_{eff}$ when the device size is varied down to the scale that does not accommodate domain wall formation:

$$j_c \propto K_{eff}/\xi \qquad (1)$$

where $\xi$ is the current-induced torque efficiency. Equation 1 suggests that an effective solution to the conflict is using materials with large $\xi$. However, as the spin Hall angle ($\theta_{SH}$), which determines the $\xi$ magnitude within the SHE model, of typical HMs such as Pt, Au, Ta, and W rarely exceeds 0.3 [3], a very high $j_c$ of order of $10^7$–$10^8$ A cm$^{-2}$ is typically needed for CIMS, which leads to device-heating and reliability concerns. The second issue is the physical origins of the in-plane current-induced torque in HM/FM/oxide heterostructures. The SHE of HM bulk and/or the SRE at interfaces are regarded as the main mechanisms of the torque origins in previous studies[7–9]. However, the existing SHE and SRE models cannot explain certain anomalous phenomena that have been observed in HM/FM/oxide heterostructures, such

as the oxygen-engineered spin–orbit torque[20–22] and the strong torque dependence on the magnetization direction[23–25], indicating that either additional mechanisms exist, or more realistic electronic structures must be considered. Realizing low $j_c$ without compromised PMA, together with understanding the physical origins of the current-induced torque in HM/FM/oxide heterostructures, is crucial for promoting practical device applications.

Here, we report the experimental observation of a very large $\xi$ of up to 2.83 and a low $j_c$ with magnitude of order of $10^6$ A cm$^{-2}$ in Pt/Co/SiO$_2$/Pt heterostructures with strong PMA ($3.92 \times 10^6$ erg cm$^{-3}$ at 300 K). The torque is anisotropic with respect to the magnetization direction and shows strong temperature dependence. We demonstrate that asymmetric orbital hybridizations at the Pt/Co and Co/SiO$_2$ play an important role in the observed phenomena, which are explained in the framework of a current-induced orbital polarization model.

## Results

**Crystal and interface electronic structures**. The ferromagnetic heterostructures studied in this work had a Si/SiO$_2$ (300)//Pt (5)/Co (1)/SiO$_2$ (1)/Pt (1) structure (the numbers in brackets correspond to the respective nominal layer thicknesses in nm). The Si/SiO$_2$ was the substrate. Figure 1a displays a high-resolution transmission electron microscopy image of the Pt/Co/SiO$_2$/Pt films. The analysis of the lattice spacings shows that the bottom Pt layer has a polycrystalline structure mainly consisting of {111} and {200} textures. The SiO$_2$ layer has an amorphous structure. We find that it is hard to discern the Pt/Co interface, which may be caused by the atom interdiffusion between the Co and Pt layers during the annealing treatment (see Methods), as Co and Pt are miscible[26]. Nevertheless, the Co and Pt layers can be distinguished by the different Z-contrasts. The elemental mappings presented in Fig. 1b provide a further confirmation of the presence of Co and Pt and more importantly, show that the Co distributes uniformly and forms a continuous film.

Figure 1c shows a high-resolution X-ray photoelectron spectroscopy spectrum of Co 2p level near the Co/SiO$_2$ interface. Peak 1 at 777.69 eV and peak 4 at 792.99 eV correspond to metallic Co$^0$ $2p_{3/2}$ and $2p_{1/2}$ levels, respectively. Peak 2 at 781.01 eV and peak 5 at 796.38 eV are assigned, respectively, to Co$^{2+}$ $2p_{3/2}$ and $2p_{1/2}$ levels. The presence of the Co$^{2+}$ at the Co/SiO$_2$ interface is due to the formation of CoO at the Co/SiO$_2$ interface during deposition of the SiO$_2$ layer[27]. Peak 3 (peak 6) lying ~6 eV above peak 2 (peak 5) is the satellite structure of peak 2 (peak 5),

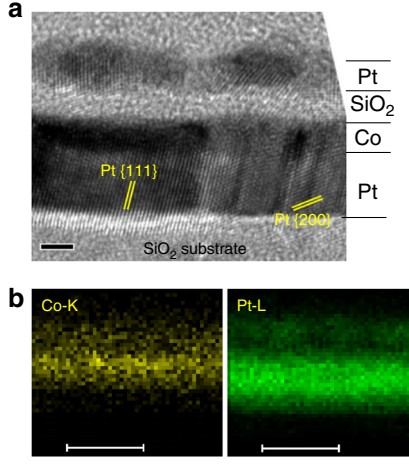

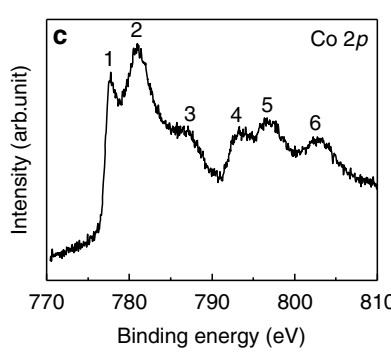

**Fig. 1** Crystal and interface-electronic structures. **a** Cross-sectional high-resolution transmission electron microscopy image of a Pt (5)/Co (1)/SiO$_2$ (1)/Pt (1) (nm) film (scale bar 2 nm). **b** Co and Pt elemental mapping obtained by energy-disperse X-ray spectroscopy (scale bar 10 nm). **c** X-ray photoelectron spectroscopy spectrum of Co 2p level near the Co/SiO$_2$ interface

which is attributed to the charge transfer from O 2p orbital to Co 3d orbital accompanying the primary photoionization process[28]. The presence of the satellite structures is a strong evidence of existence of the Co 3d−O 2p orbital hybridization at the Co/SiO$_2$ interface.

**Magnetic properties**. Figure 2a, b show the magnetic hysteresis (MH) and anomalous Hall effect (AHE) resistance ($R_{AHE}$) loops measured at temperatures of 50, 300, and 400 K with the external magnetic field $\mathbf{H}^{ext}$ perpendicular to the film plane. These loops present sharp magnetization switching as the magnitude of $\mathbf{H}^{ext}$ ($H^{ext}$) changes and have high remanence, indicating that the films have well-established PMA, which is due to the Pt 5d−Co 3d orbital hybridization at the Pt/Co interface[1] and the Co 3d−O 2p orbital hybridization at the Co/SiO$_2$ interface[2]. As will be presented below, the combined effect of the Pt 5d−Co 3d−O 2p orbital hybridization can produce a large orbital splitting in the Co layer, which not only stabilizes the PMA state but also induces a giant torque on the Co magnetization with current injection into the films.

The temperature dependence of the saturation magnetization ($M_s$), $R_{AHE}$ and coercivity field ($H_c$) of the MH and $R_{AHE}$ loops are shown in Fig. 2c, d. All these quantities increase with decreasing temperature, typical phenomena of ferromagnetic heterostructures. We notice that there is a difference in $H_c$ between the MH and $R_{AHE}$ loops, which we ascribed to the different sample size for MH and AHE measurements, the microfabrication process, and the measurement technique used (see Methods).

Ultrathin metallic FM layers with thickness <1.5 nm usually exhibit a granule-like morphology when grown on an oxide layer[29,30], because the difference in the surface-free energy ($\delta$) between the metal and oxide is typically large[31,32]. It was demonstrated that the granule-like morphology can have significant impact on the evaluation of the spin-transfer-torque switching efficiency[30]. However, the difference in $\delta$ between Co and Pt is very small ($\delta_{Co} = 2.71\,\mathrm{J\,m^{-2}}$ and $\delta_{Pt} = 2.69\,\mathrm{J\,m^{-2}}$, ref. [31]); the Co is expected to form a continuous film on the Pt

even at a small thickness of 1 nm, supported by the element mapping shown in Fig. 1b. Moreover, no multistep-like switching and/or superparamagnetic behavior are found in the MH and $R_{AHE}$ curves, further confirming that the Co layer is morphologically continuous and magnetically homogeneous. Therefore, we believe that the granular effect has little influence on the $\xi$ evaluation below.

**Evaluation of torque efficiency and magnetic anisotropy**. When a charge current is injected along the film plan of HM/FM/ oxide heterostructures, the FM magnetization can experience a torque, which is tentatively attributed to the spin polarizations generated by the bulk SHE and/or the interface SRE in previous studies[7–9]. Although the exact origin of the in-plane current-induced torque in HM/FM/oxide structures remains debatable, a number of experimental and theoretical studies[12,18,23–25,33] have demonstrated that it can separate into two orthogonal components regardless of detailed mechanisms. One is an even function of the magnetization expressed as $\mathbf{T}^{AD} = T^{AD}\mathbf{m} \times (\mathbf{\sigma} \times \mathbf{m})$ and is called the antidamping torque, because it can compensate the intrinsic Gilbert magnetic damping and induces magnetization switching. The other one is an odd function of the magnetization expressed as $\mathbf{T}^{FL} = T^{FL}(\mathbf{m} \times \mathbf{\sigma})$ and is called the field-like torque, as it can induce magnetization precession like a magnetic field does. Here, $\mathbf{m}$ is the magnetization unit vector, $\mathbf{\sigma}$ is the current-induced angular momentum polarization and $T^{AD}$ and $T^{FL}$ describe the magnitudes of $\mathbf{T}^{AD}$ and $\mathbf{T}^{FL}$, respectively.

To quantify the two torque components, we measured $R_{AHE}$ as a function of $H^{ext}$ with $\mathbf{H}^{ext}$ slightly tilting with respect to the xy-plane (i.e., almost along the hard magnetization axis, see the insets of Fig. 3a, b). This measurement scheme was introduced to ensure that the magnetization can rotate coherently without domain formation, such that a macrospin method is applicable to investigate the interplay of $\mathbf{T}^{AD}$ and the torques exerted by $\mathbf{H}^{ext}$ and the anisotropy field $\mathbf{H}^{an}$ (see below). Figure 3a, b show the normalized $R_{AHE}$ (i.e., cosθ) curves obtained for a Pt/Co/SiO$_2$/Pt device with positive (along $\mathbf{y}$) and negative (along $-\mathbf{y}$) currents,

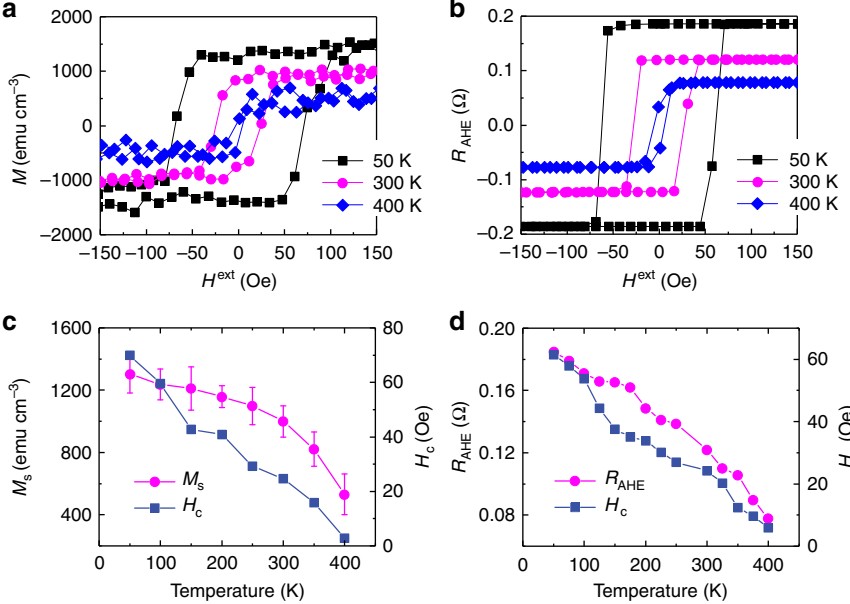

**Fig. 2** Temperature-dependent magnetic properties. **a** Magnetization (M) and **b** $R_{AHE}$ as a function of $H^{ext}$ for Pt/Co/SiO$_2$/Pt heterostructures with $H^{ext}$ perpendicular to the film plane and measurement temperature of 50, 300, and 400 K. **c, d** Temperature dependence of $M_s$, $R_{AHE}$, and $H_c$. The error bars in **c** are defined as s.d. The $H_c$ in (**c**) is obtained from M–$H^{ext}$ curve and that in (**d**) is extracted from $R_{AHE}$–$H^{ext}$ curve

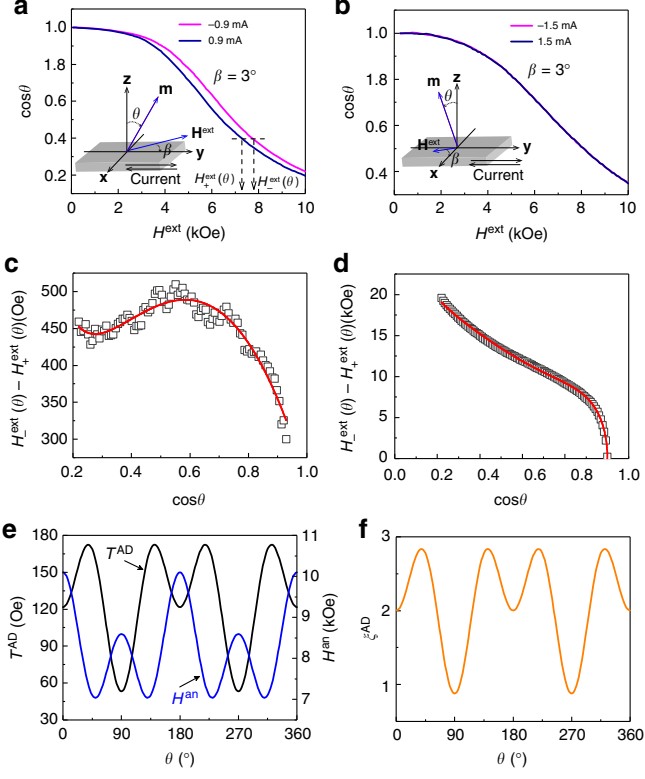

**Fig. 3** Quantification of current-induced torque and magnetic anisotropy. Normalized $R_{AHE}$ (i.e., $\cos\theta$) as a function of $H^{ext}$ for positive (along **y**) and negative (along −**y**) currents, when $H^{ext}$ is in (**a**) in the $yz$-plane and **b** in the $xz$-plane. Insets of (**a**) and **b** show the measurement scheme. The curves were obtained at 300 K and $\beta = 3°$. **c** Values of $H_+^{ext}(\theta) - H_-^{ext}(\theta)$ determined from the data in (**a**) (squares) and best-fit curve obtained using Eq. 2 (solid line). **d** Values of $H_+^{ext}(\theta) + H_-^{ext}(\theta)$ determined from the data in **a** (squares) and best-fit curve obtained using Eq. 3 (solid line). **e** Dependences of $T^{AD}$ (black line) and $H^{an}$ (blue line) on $\theta$. **f** $\xi^{AD}$ as a function of $\theta$

measured at 300 K and $\beta = 3°$. Here, $\theta$ is the angle between **m** and the film normal **z**, and $\beta$ is the angle between $H^{ext}$ and the $xy$-plane. The $R_{AHE}$ decreases slowly with increasing $H^{ext}$, which is due to the coherent rotation of the Co magnetization toward the hard magnetization axis. According to the torque symmetries, the $R_{AHE}$ curves measured with $H^{ext}$ in the $yz$-plane reflects mostly $\mathbf{T}^{AD}$ contributions, whereas those measured with $H^{ext}$ in the $xz$-plane reflects mostly $\mathbf{T}^{FL}$ terms[9,23]. When $H^{ext}$ sweeps in the $yz$-plane (Fig. 3a), significant curve splitting is apparent for currents of ±0.9 mA, indicating that $\mathbf{T}^{AD}$ is sizeable. However, no curve splitting is visible when $H^{ext}$ is in the $xz$-plane (Fig. 3b), even for currents of ±1.5 mA, suggesting that $\mathbf{T}^{FL}$ is negligible. We therefore focus only on $\mathbf{T}^{AD}$ below.

We next extract $T^{AD}$ and the magnitude of $H^{an}$ (i.e., $H^{an}$) using a macrospin method[8,9,12,23–25]. When considering the space symmetry of torques[23], $\mathbf{T}^{DL}$ takes the form $\mathbf{T}^{AD} = \mathbf{m} \times (\boldsymbol{\sigma} \times \mathbf{m}) T_0^{AD} + (\mathbf{z} \times \mathbf{m})(\mathbf{m}\cdot\mathbf{y})[T_2^{AD} + T_4^{AD}(\mathbf{z} \times \mathbf{m})^2]$, where $T_0^{AD}$, $T_2^{AD}$, and $T_4^{AD}$ correspond to the zeroth-, second-, and fourth-order terms in $T^{AD}$, respectively. It is straightforward to have $T^{AD} = T_0^{AD} + T_2^{AD}\sin^2\theta + T_4^{AD}\sin^4\theta$. In addition, the uniaxial MCA has a corresponding $H^{an}$ with form of $H^{an} = H_0^{an} + H_2^{an}\sin^2\theta + H_4^{an}\sin^4\theta$, where $H_0^{an}$, $H_2^{an}$, and $H_4^{an}$ represent the zeroth-, second-, and fourth-order terms in $H^{an}$, respectively. Without loss of generality, we consider the lowest- and high-order terms of $T^{AD}$ and $H^{an}$ in the macrospin method simultaneously. From

Supplementary Note 1, we have

$$H_-^{ext}(\theta) - H_+^{ext}(\theta) = 2(T_0^{AD} + T_2^{AD}\sin^2\theta \\ + T_4^{AD}\sin^4\theta)/\cos(\theta + \beta) \tag{2}$$

$$H_-^{ext}(\theta) + H_+^{ext}(\theta) = 2(H_0^{an} + H_2^{an}\sin^2\theta \\ + H_4^{an}\sin^4\theta)\cos\theta\sin\theta/\cos(\theta + \beta) \tag{3}$$

where $H_+^{ext}(\theta)$ and $H_-^{ext}(\theta)$ are the $H^{ext}$ values that produce the same $\theta$ for positive and negative currents, respectively. Taking the difference between $H_+^{ext}(\theta)$ and $H_-^{ext}(\theta)$ (Fig. 3c) and fitting the resulting data to Eq. 2, we obtain $T_0^{AD} = 121.7$ Oe, $T_2^{AD} = 256.9$ Oe, and $T_4^{AD} = -325.3$ Oe for a current of 0.9 mA. Evidently, $T^{AD}$ contains substantial high-order terms $T_2^{AD}$ and $T_4^{AD}$, even larger than the zeroth-order term $T_0^{AD}$, which suggests that $T^{AD}$ presents a strong anisotropy. The result of the anisotropic $T^{AD}$ is in contrast to previous studies where $T^{AD}$ was demonstrated to be angular-independent[8,9]. We temporarily ascribed such a diversity to the difference in the electronic structures of films (see Supplementary Note 7). Figure 3d shows the sum of $H_+^{ext}(\theta)$ and $H_-^{ext}(\theta)$ as a function of $\cos\theta$. By fitting the curve to Eq. 3, $H_0^{an}$, $H_2^{an}$, and $H_4^{an}$ were estimated to be 10.09, −10.47, and 8.97 kOe, respectively. Using the formula $K_u = K_1\sin^2\theta + K_2\sin^4\theta + K_3\sin^6\theta$, where $K_u$ is the uniaxial magnetic anisotropy energy and $K_1$ (=$H_0^{an}M_s/2$), $K_2$ (=$H_2^{an}M_s/4$), and $K_3$ (=$H_4^{an}M_s/6$) are the anisotropy energy constants, $K_{eff}$ (i.e., the $K_u$ value at $\theta = 90°$) was estimated to be $3.92 \times 10^6$ erg cm$^{-3}$; this value is comparable to that of the perpendicularly magnetized Co/Pt multilayers[26] but considerably larger than those of Ta/CoFeB/MgO[8] and W/CoFeB/MgO[12].

Having obtained the coefficients $T_n^{AD}$ and $H_n^{an}$ ($n = 0, 2$, and 4), we investigate the detailed angular distributions of $T^{AD}$ and $H^{an}$ based on their space symmetries and, more importantly, the possible interrelation between the two quantities. The variation of $T^{AD}$ and $H^{an}$ as a function of $\theta$ is plotted in Fig. 3e. Interestingly, $T^{AD}$ and $H^{an}$ exhibit an opposite $\theta$ dependence, i.e., when $H^{an}$ decreases (increases) with $\theta$, $T^{AD}$ increases (decreases), pointing to a common mechanism which links up $T^{AD}$ and $H^{an}$.

The $T^{AD}$ efficiency $\xi^{AD}$ can be estimated using the formula (modified from Pai et al.[34] and Khvalkovskiy et al.[35])

$$\xi^{AD} = (2e/\hbar)M_s t_{Co}^{eff}(T^{AD}/j_{PtCo}) \tag{4}$$

Here, $e$ is the electron charge; $\hbar$ is the reduced Planck constant; $t_{Co}^{eff} = 0.72$ nm is the effective Co layer thickness (see Supplementary Note 3); and $j_{PtCo} = 1.35 \times 10^6$ A cm$^{-2}$ is the current density shunting in the Pt (5)/Co (1) (nm) bilayer for a current of 0.9 mA (see Supplementary Note 4). It is evident that $\xi^{AD}$ is anisotropic as shown in Fig. 3f: it increases from 2.0 at $\theta = 0°$ to 2.83 at $\theta \sim 40°$, and then decreases to 0.88 as $\theta$ is further increased to 90°. These $\xi^{AD}$ values are significantly higher than those previously reported in HM/FM/oxide structures such as (Pt, Pd, Ta, W)/(Co, CoFe, CoFeB)/(AlO$_x$, MgO)[8,9,12,23,25,34], which rarely exceeds 0.3.

Note here that the increase in device temperature due to current-induced Joule heating was negligibly small for a 0.9-mA current injection (see Supplementary Note 5), suggesting that thermoelectric effects such as anomalous Nernst and spin Seebeck effects should have little influence on the evaluated $\xi^{AD}$ values. Previous studies[36] have demonstrated that the thermoelectric effects are very small in Pt/Co systems. Additionally, the planar Hall effect was found to be two orders of magnitude smaller than the AHE in our Pt/Co/SiO$_2$/Pt films (Supplementary Note 2) and thus should not influence upon the $\xi^{AD}$ estimation.

**Temperature dependences of current-induced torque and magnetic anisotropy.** We also measured the $R_{AHE}$–$H^{ext}$ curves with $\beta = 3°$ at different temperatures to study the temperature dependences of $T^{AD}$ and $K_u$, which are shown in Fig. 4a, b respectively. The lowest-order terms $T_0^{AD}$ and $K_1$ increase with decreasing temperature. With regard to the high-order terms, $T_2^{AD}$ and the absolute value of $T_4^{AD}$ decrease with decreasing temperature; the absolute value of $K_2$ shows a not strictly monotonic increase with decreasing temperature and presents a salient at ~125 K, while $K_3$ have a monotonic temperature dependence. Despite the inverse scaling of high-order terms of $T^{AD}$ and $H^{an}$ with temperature, the $K_{eff}$ and the degree of anisotropy of $T^{AD}$ (i.e., $|T_2^{AD} + T_4^{AD}|$) were found to possess similar temperature dependence (Fig. 4c); they increase with decreasing temperature, suggesting that the anisotropic $T^{AD}$ may have the same origin as $K_u$.

**CIMS measurements.** To examine the CIMS, we applied a small, fixed $H^{ext}$ along **y** or $-$**y**, i.e., $\beta = 0°$ or 180°, and then swept the quasistatic direct current with a ramp rate of 0.5 mA s$^{-1}$. The CIMS experiment was performed at 300 K and monitored by measuring $R_{AHE}$. The application of the small $H^{ext}$ was to break the time reversal symmetry in the $xy$-plane so that deterministic CIMS can be achieved[37]. We found that the $H^{ext}$ with magnitude of 25–400 Oe tilted the average magnetization by ~0.5–3.5° from **z** (Supplementary Note 6), but did not provide any preference for either up or down magnetization state in the absence of current injection. In Fig. 5a, where the $H^{ext}$ is applied along **y**, sweeping the quasistatic direct current results in hysteretic magnetic switching, where the magnetization states depend on the current direction. Moreover, the switching curves reverse their polarity when the $H^{ext}$ direction is changed (Fig. 5b), a characteristic of $\mathbf{T}^{AD}$-induced magnetization switching[7–9]. Similar to the field-driven switching (Fig. 2a, b), the current-driven loops shown here also exhibit sharp magnetization switching, which is due to the domain wall nucleation and propagation.

The $R_{AHE}$ of the curves with $H^{ext} = \pm 25$ Oe is ~0.045 Ω; this value is much smaller than that of the full-switching curve at 300 K shown in Fig. 2b, which indicates that, in this case, there is partial magnetization switching. For the curves with $H^{ext} = \pm 50$ Oe, complete switching is observed, with an average $j_c = (|j_c^{up}| + |j_c^{down}|)/2 \approx 5.83 \times 10^6$ A cm$^{-2}$. Here, $j_c^{up}$ and $j_c^{down}$ denote the $j_c$ values at which the magnetization switches from down to up and up to down, respectively. As $H^{ext}$ is further increased to $\pm 400$ Oe, the average $j_c$ decreases to ~$2.53 \times 10^6$ A cm$^{-2}$. These $j_c$ values are much smaller than those previously reported in Pt-based HM/FM/oxide heterostructures (usually in the order of $10^7$ A cm$^{-2}$, see refs. [7,9,20]).

Note that the macrospin method is based on probing the static magnetization whereas the CIMS is by the dynamic domain wall formation and propagation. Discrepancy between the static and dynamic torque may be expected, because there can be an enormous number of pathways to overcome the energy barrier for domain-wall-based magnetization switching[38]. Therefore, the large $\xi^{AD}$ is not the exclusive causal factor for the low $j_c$ values obtained here. To analyze the effects of the current-induced torque on dynamic magnetization quantitatively, a better approach is to investigate the temporal information of the torque by measuring the lifetime of magnetization reversal under the presence of an electric current[38–40].

## Discussion

The SHE of HM bulk and the interfacial SRE are two possible mechanisms of in-plane current-induced torque in HM/FM/oxide heterostructures. In the SHE interpretation, spin currents

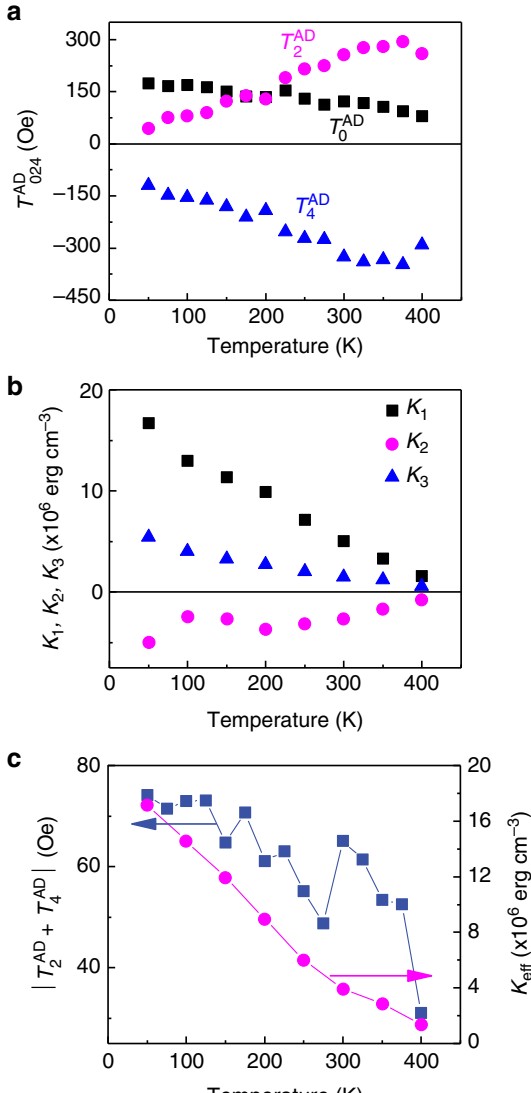

**Fig. 4** Temperature effects on the current-induced torque and magnetic anisotropy. **a** Temperature dependence of the zeroth- ($T_0^{AD}$), second-($T_2^{AD}$), and fourth-order ($T_4^{AD}$) terms in $T^{AD}$. **b** Temperature-dependent $K_1$, $K_2$, and $K_3$. **c** Absolute values of $T_2^{AD} + T_4^{AD}$ and $K_{eff}$ as a function of temperature

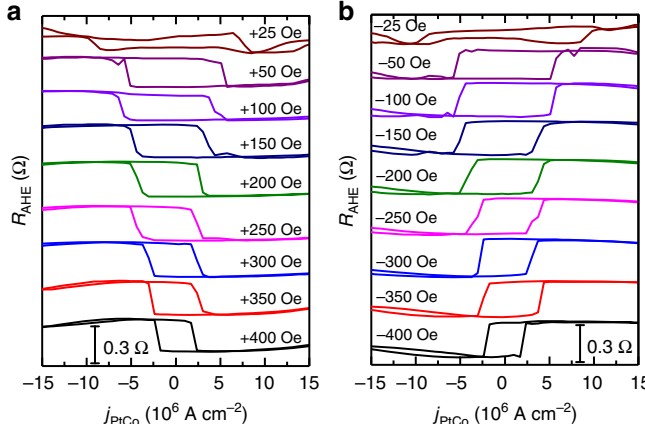

**Fig. 5** CIMS measurements. $R_{AHE}$ as a function of $j_{PtCo}$ passing through the Pt (5)/Co (1) (in nm) bilayer in the presence of a variable $H^{ext}$ along (**a**) $-$**y**- and (**b**) $-$**y**-direction. The measurement was performed at ~300 K

generated in the HM bulk diffuse into the adjacent FM layer and exerts torques upon the FM magnetization. However, theoretical calculations[33] have predicted SHE-generated $T^{AD}$ not to have any magnetization-direction dependence; the $\theta_{SH}$ of Pt bulk was suggested theoretically and experimentally to be less than 0.1 [41,42], which is too small to be accounted for the large $\xi^{AD}$ obtained here. Moreover, reported experimental works have demonstrated that the intrinsic SHE is almost unchangeable with temperature[43,44]. Therefore, the SHE interpretation is inadequate to explain the giant, anisotropic, and strongly temperature-dependent $T^{AD}$ and $\xi^{AD}$ observed here.

On the other hand, theoretical calculations[45] based on the SRE model including the anisotropic D'yakonov–Perel spin relaxation predicted that $T^{AD}$ will acquire an angular dependence when the Rashba SOC energy is considerably larger than the exchange interaction. However, this analytically tractable condition may be untenable in our work, because the Rashba effect and exchange interaction are both large in our films, as hinted by the presence of the strong PMA[46]. In addition, the absence of $T^{FL}$ in our study suggests that the extrinsic scattering-induced spin relaxation,

which typically generates $T^{FL}$ in the context of SRE model[33,45], is not responsible for the anisotropic $T^{AD}$.

Our results suggest that additional effects contribute to $T^{AD}$. It is well known that the anisotropy of the $3d$ orbital moment (OM) is the microscopic origin of MCA in $3d$ FM[47]. Learning from the close relationship between $T^{AD}$ and MCA and the strong temperature dependence of $T^{AD}$ shown above, we infer that the intrinsic $3d$ OM anisotropy is probably a key to understand our results. Particularly, orbital polarization $\Delta L$ in response to an electric current has recently been predicted in nonmagnetic tellurium crystals lacking bulk inversion symmetry[48]. Here, $L$ is the orbital angular momentum operator. In principle, such a current-induced $\Delta L$ should also exist in our Pt/Co/SiO$_2$/Pt films with symmetry-breaking interfaces. There, the Pt $5d$–Co $3d$ and Co $3d$–O $2p$ orbital hybridizations modify the charge distribution and give rise to a strong interfacial crystal electric field ($E_{CEF}$) at the respective interface (Fig. 6a). Because the Co $3d$ orbitals are exposed to the crystal environment, the interfacial $E_{CEF}$ primarily splits the $3d$ orbitals into two states with opposite OM (L and −L), while the SOC, which is not strong in Co metal, only acts as

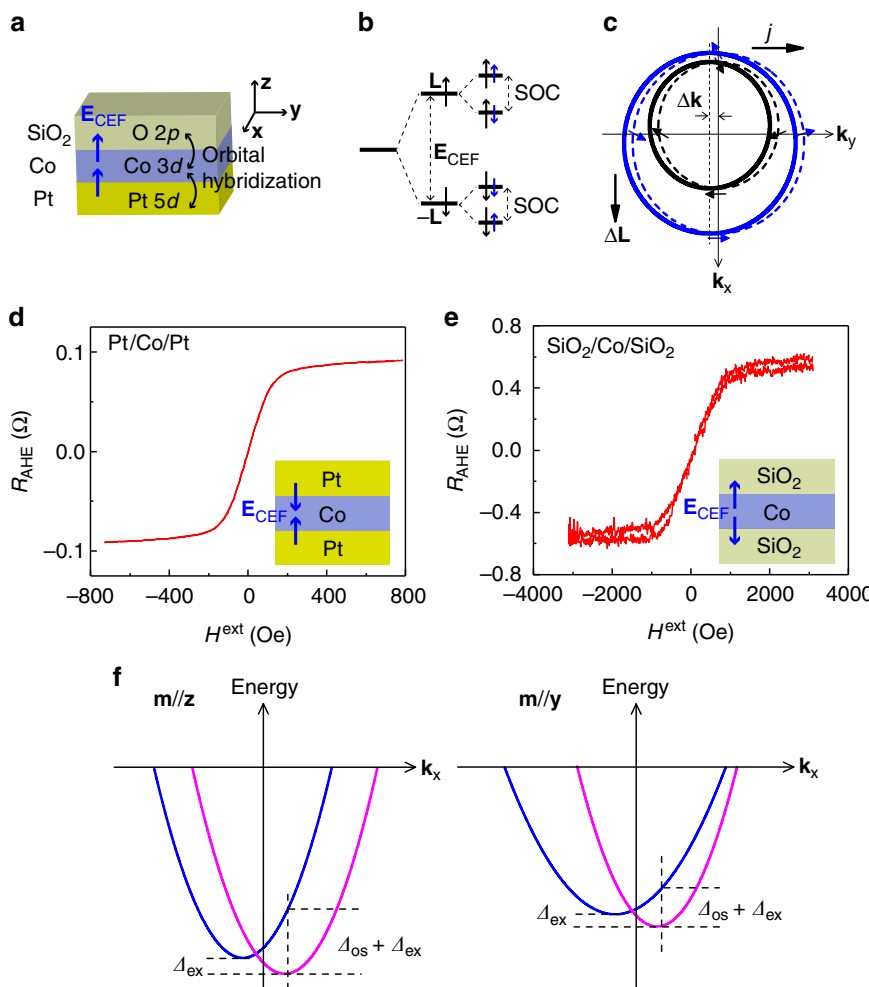

**Fig. 6** Schema of the OREE. **a** Pt $5d$–Co $3d$ and Co $3d$–O $2p$ orbital hybridizations at the Pt/Co and Co/SiO$_2$ interfaces produce a $E_{CEF}$ at the respective interface. **b** Energy scales for illustration of how a degenerate state becomes split by $E_{CEF}$ and SOC. Here, the black and blue arrows represent the orbital and spin moments, respectively. **c** Sketch of simplified Rashba-like chiral orbital textures at the Fermi surface. The dashed black and blue circles represent the inner and outer Fermi contours with equilibrium, i.e., without current injection, respectively. The arrows indicate the orbital moments. A current density ($j$) injection along **y** shifts the outer and inner Fermi contours along −**y** (i.e., $\Delta k$), producing a lateral $\Delta L$. **d**, **e** $R_{AHE}$-$H^{ext}$ curves for the controlled samples Pt (5)/Co (1)/Pt (1)/SiO$_2$ (1)/Pt (1) (in nm, short for Pt/Co/Pt) and Pt (5)/SiO$_2$ (5)/Co (1)/SiO$_2$ (1)/Pt (1) (in nm, short for SiO$_2$/Co/SiO$_2$). The insets show the interfacial $E_{CEF}$ configurations of the two controlled samples. **f** Sketches of Rashba-type band structures for **m**||**z** and **m**||**y** to illustrate how the magnetization direction-dependent orbital torque arises

a perturbation to the band energy governed by the orbital splitting and further splits them into states of spin (Fig. 6b). The split orbital states couple with their own momentum $\hbar\mathbf{k}$ via $\mathbf{E}_{CEF}$[49], i.e., $\mathbf{E}_{CEF} = \mathbf{L} \times \hbar\mathbf{k}$, forming Rashba-like chiral orbital textures in $\mathbf{k}$-space shown in Fig. 6c, where the dashed circles represent the Fermi contours at equilibrium and $\mathbf{k}$ is the electron wave vector. When an in-plane current passes through the Co layer, the shift of the Fermi contours (solid circles in Fig. 6c) produces an asymmetric electron distribution in $\mathbf{k}$-space and leads to a local nonequilibrium $\Delta\mathbf{L}$ with polarization direction perpendicular to the current path. Contrast to the nonmagnetic tellurium crystals with bulk inversion asymmetry, the current-induced $\Delta\mathbf{L}$ in our films is interface-originated and can further couple to and exert torques upon the Co magnetization via exchange interaction. Therefore, we dub this magnetoelectric effect as the orbital Rashba–Edelstein effect (OREE) and refer to the corresponding torque as orbital torque to underline its orbital origin.

We next discuss the important role of asymmetric orbital hybridizations of the Pt/Co and Co/SiO$_2$ interfaces in achieving large orbital splitting and the giant $\xi^{AD}$ we observed. As already mentioned, the spin states are dominated by the orbital splitting and, thus, in turn, can be taken as a gauge of the size ($\Delta_{os}$) of the orbital splitting. In the Pt/Co/SiO$_2$/Pt films, the interfacial $\mathbf{E}_{CEF}$ of the Co/SiO$_2$ interface increases in passing from Co to SiO$_2$, and that of the Pt/Co interface increases in passing from Pt to Co[50]; the cumulative effect is a significant enhancement of the total interfacial $\mathbf{E}_{CEF}$ that favors a large $\Delta_{os}$, which overwhelms the demagnetization field and stabilizes the spin states at the normal direction, causing the PMA of the films (Fig. 2a, b). To make this point clearer, we investigated the magnetic properties of two controlled samples Pt (5)/Co (1)/Pt (1)/SiO$_2$ (1)/Pt (1) (thickness in nm, Pt/Co/Pt for short) and Pt (5)/SiO$_2$ (5)/Co (1)/SiO$_2$ (1)/Pt (1) (thickness in nm, SiO$_2$/Co/SiO$_2$ for short). The AHE loops in Fig. 6d, e show that the two controlled samples have in-plane magnetic anisotropy, as expected. This is because that the interfacial $\mathbf{E}_{CEF}$ of the Pt/Co (SiO$_2$/Co) and Co/Pt (Co/SiO$_2$) interfaces has opposite direction and tends to cancel out (insets of Fig. 6d, e), weakening the $\Delta_{os}$ and thereby leading to the in-plane magnetic anisotropy. As the $\Delta_{os}$ also dominates the amplitude of the current-induced $\Delta\mathbf{L}$, we believe that a pronounced Co 3$d$ orbital splitting due to the combined effect of asymmetric orbital hybridizations at the Pt/Co and Co/SiO$_2$ interfaces is the key element for the giant $\xi^{AD}$ that was obtained in our work.

In magnetized systems, the band structures depend not only on the intrinsic $\Delta_{os}$ but also on the magnetization direction, due to exchange field-induced Fermi surface distortion effect. Figure 6f sketches Rashba-type band structures for $\mathbf{m}||\mathbf{z}$ and $\mathbf{m}||\mathbf{y}$ to illustrate how the magnetization direction-dependent orbital torque arises. Generally, the exchange field shifts the two Rashba-type subbands in opposite directions along the energy-axis, causing an exchange splitting ($\Delta_{ex}$). The total energy splitting $\Delta_{tot}$ for a given $\mathbf{k}$ is then determined by both $\Delta_{os}$ and $\Delta_{ex}$, i.e., $\Delta_{tot} = \Delta_{os} + \Delta_{ex}$. For $\mathbf{m}||\mathbf{z}$, the exchange field have no distortion effect on the subband structures. As the magnetization rotates toward $\mathbf{y}$, an in-plane component of the exchange field appears, which shifts the two Rashba subbands in opposite directions along the $\mathbf{x}$-axis and distorts the Fermi surfaces. It is apparent that the shift and distortion of the Fermi surfaces depend on the size of the in-plane component of the exchange field, i.e., they are dependent on the magnetization direction. As a result, the $\Delta_{tot}$ becomes angular-dependent, leading to the anisotropic $\xi^{AD}$ that was observed.

The temperature dependence of the orbital torque can be interpreted as a phonon-mediated electron hopping effect. As the temperature is increased, the atomic vibration becomes stronger and promotes electron hopping between hybridizing Pt 5$d$, Co 3$d$, and O 2$p$ orbitals via electron–phonon interaction. It is expected

that the electron occupations in the hybridized orbitals become more symmetrical upon increasing temperature, which weakens the interface $\mathbf{E}_{CEF}$ and reduces the $\Delta_{os}$, resulting in the decrease of the orbital torque anisotropy (Fig. 4c). The different temperature dependence of $T_0^{AD}$, $T_2^{AD}$, and $T_4^{AD}$ (Fig. 4a) may originate from the different temperature dependence of the lowest- and high-order terms in $\mathbf{E}_{CEF}$. This is supported by the dramatic temperature dependence of $K_1$, $K_2$, and $K_3$ (Fig. 4b), which has been well studied within a temperature-dependent crystal field model[51]. To our knowledge, however, the temperature-dependent high-order terms in $T^{AD}$ has not been studied systematically yet. To get further insights into it, more experiments and calculations based on realistic electronic structures are needed.

In summary, we have reported a large, anisotropic, and strongly temperature-dependent current-induced antidamping orbital torque in Pt/Co/SiO$_2$/Pt heterostructures with high PMA. The results cannot be simply understood using the existing SHE and SRE but can be qualitatively interpreted within an OREE model we proposed, where the antidamping torque is related to intrinsic Co 3$d$ band structures and does not require transport of the current-induced nonequilibrium polarizations. The asymmetric orbital hybridizations at Pt/Co and Co/SiO$_2$ interfaces can enhance the orbital splitting significantly and lead to a very large torque efficiency, making the orbital torque useful for potential applications in high-performance magnetic devices. Our results highlight the critical role of the orbital anisotropy in the generation of in-plane current-induced torque in HM/FM/oxide structures, and may improve the understanding of the previously observed anomalous spin–orbit torque phenomena[20–25]. In addition, the multiplicity of the orbital symmetries and their sensitivity to the interfacial orbital hybridization provide great opportunities to optimize the torque efficiency and switching current, for example, by orbital engineering using electric-field-driven ionic migration effects[52–54].

## Methods

**Sample deposition and device fabrication.** All films studied here were deposited on Si/SiO$_2$ (300 nm) substrates by a magnetron sputtering system at room temperature. The Pt and Co were deposited by direct-current sputtering and the SiO$_2$ were deposited by radio-frequency sputtering. The deposition rates of Pt, Co, and SiO$_2$ were 0.08, 0.04, and 0.02 nm s$^{-1}$, respectively. The base pressure of the sputtering system was better than $3 \times 10^{-7}$ Torr and the working argon pressure was 4 mTorr. After deposition, the films were fabricated into Hall devices using ultraviolet lithography and ion beam etching. During the microfabrication process, the films were heated to 100 °C to solidify the photoresist and remove the deionized water at surface. The device channel width $w$ is 10 μm, and the length $l$ between two neighboring Hall probes is 80 μm. Au (50 nm)/Cu (10 nm) were used as contact electrodes, which were made by direct-current sputtering and lift-off technology. To improve the perpendicular magnetic anisotropy, all devices were annealed at 200 °C for 30 min in a vacuum furnace with a base pressure of $3 \times 10^{-7}$ Torr. Two Pt films with thicknesses of 5 and 1 nm were also deposited separately and fabricated into Hall devices to determine their resistivity.

**Electrical and magnetic measurements.** The devices were placed on a sample rotator that can perform 360° rotation with a 0.05° precision for electrical measurements. A Keithley 2400 sourceMeter was used for quasistatic direct current sweeping and a Keithley 2182A nanovoltmeter was used for voltage acquisition. The ramp rate of the sweeping current was 0.5 mA s$^{-1}$. The room-temperature resistivity of the Pt (5 nm), Co (1 nm), and Pt (1 nm) layers was determined to be 55.6, 37.6, and 80.1 μΩ cm, respectively (see Supplementary Note 4). Samples with an area of $3 \times 4$ mm$^2$, annealed at 200 °C for 30 min, were used for magnetization hysteresis loop measurements using a vibrating sample magnetometer (VSM). Note here that the sample area in electrical and magnetic measurements is different.

**X-ray photoelectron spectroscopy measurements.** The base pressure of the X-ray photoelectron spectroscopy (XPS) system (Thermo Scientific Escalab 250Xi) was better than $5 \times 10^{-8}$ Pa. The source of X-rays was Al $K_\alpha$. The energy analyzer was operated at constant pass energy of 30 eV. The samples used for XPS measurements had an area of $10 \times 10$ mm$^2$ and were annealed at 200 °C for 30 min. Before loading into the XPS system chamber, the samples were cleaned with acetone using an ultrasonic apparatus to remove surface contaminants. To

eliminate the charge effect, all binding energies were calibrated by the Pt $4f_{7/2}$ level (70.9 eV).

**High-resolution transmission electron microscopy**. Cross-sectional specimens for high-resolution transmission electron microscopy (HRTEM) was prepared by mechanical grinding, polishing and dimpling, followed by Ar-ion milling using a Gatan 695 precision ion polishing system. The HRTEM images and element mappings were acquired through a FEI Technai F20 equipped with an energy dispersive X-ray detector. The accelerating voltage for operation is 200 kV.

**Data availability**. All data needed to evaluate the conclusions in the paper are present in the paper and/or the Supplementary Information. Additional data related to this paper are available from the corresponding author on request.

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

## Acknowledgements

This work was financially supported by the National Key Basic Research Program of China (2014CB931702), the China Postdoctoral Science Foundation (AD41712), NSFC (61725402, 21403109), the Fundamental Research Funds for the Central Universities (Nos. 30915012205 and 30916015106), Natural Science Foundation of Jiangsu Province (BK20140769), PAPD of Jiangsu Higher Education Institutions.

## Author contributions

X.C. and H.B.Z. conceived and designed the experiments. X.C. fabricated the samples and performed the electrical and magnetic measurements with the help of G.Y. and H.S. X.C., G.Y., and Y.L. analyzed the data. X.C., Y.L., C.H., M.H.L., and H.B.Z. wrote the manuscript. All authors discussed the results and commented on the manuscript. H.B.Z. supervised the project.

## Additional information

**Competing interests:** The authors declare no competing interests.

