## [Peer Review File · Nature Communications]

Reviewers' comments:

Reviewer #1 (Remarks to the Author):

The paper discusses Spin-Orbit-Torque switching of a magnetically perpendicular Co film sandwiched between Pt and SiO₂. The authors find low critical current density for switching and discuss the results within a macrospin-model. They evaluate the magnetic anisotropy field and the torques within this model. The two main results are a high torque efficiency and an anisotropic torque.

While the paper is well written, the scientific content is not novel enough and the conclusions are not unambiguous enough to justify publication in Nature Communications.

The main criticisms are:

1. The authors claim for their critical current density “These j-c values are much smaller than those obtained through the SHE mechanism ...”. It is true, that the j-c values shown in this paper are relatively low. Similar values, however, have been already published (see, e.g., Appl. Phys. Lett. 109, 142405 (2016)). Moreover, the samples discussed in this manuscript cannot be directly compared with those of other publications. A look into ref. 11 of the manuscript (Phys. Rev. Lett. 109, 096602 (2012)) shows, that the samples discussed in the PRL (Pt/Co/Al₂O₃) exhibit a coercive field of more than 150 Oe, while the samples discussed in the present manuscript have only around 25 Oe. That means, that the effective anisotropy, which determines the switching current density is much smaller in the samples discussed in the manuscript than those of, e.g., ref. 11.

2. In contrast with this apparent coercive field H-c, the authors evaluate an anisotropy field H-an varying between 7 kOe and 10 kOe depending on the angle between magnetization and z-axis. When the magnetization is perpendicular, then H-an is 10 kOe, i.e. almost three orders of magnitude larger than H-c.

In such cases, the magnetization switching is obviously governed by thermal activation and arbitrarily low critical current densities for SOT switching can be achieved if the current flows for a long time. In other words, the values of j-c will depend strongly on the pulse duration as well as on the temperature with j-c going to zero as T approaches the temperature, where H-c becomes zero (around 400K in this case, see Figure 3b of the manuscript). Such thermally activated behaviour was already discussed in the literature for CoFeB/MgO based tunnel junctions (see Timopheev, A.A. et al., Phys. Rev. B 96, 014412 (2017)) and attributed to a crossover from a homogeneous film to a granular type. There, the magnetization dynamics is

governed by small activation energies of granules instead of the anisotropy energy of the complete film. Therefore, the main conclusion of the paper, e.e. a large value of the torque efficiency is highly questionable.

3. While the discussion of the critical current density is, therefore, somehow meaningless, the evaluation of the anisotropy and the torques is within a macrospin model and has scientific value. In particular the finding of an anisotropic torque seems interesting. There are, however, some questions arising:

3.1 In Figure 1b, there is a substantial difference between the M(H) loop and the loop from the anomalous Hall effect. This cannot be true if the same sample was used for the measurement. I suspect, that the processing of the M(H) data (cancelling of para- or diamagnetic background) is the reason.

3.2 The temperature, at which the experiments have been done must be given for all results, because both the coercive fields as well as the critical switching currents strongly depend on temperature.

3.3 To understand the data, it is necessary to also give the shape of the current pulses that have been driven through the sample.

3.4 The control samples are not really of value for the paper, because the discussion is not deep enough. Do the authors really suggest, that a sample such as Pt/Co/Pt has no contribution of the interfaces that favour a PMA at small Co thickness? This would be in contrast to all experimental observations (see, e.g., J. Magnet. Magnet. Mater. 93, 194 (1991)).

3.5 In Figure 2a, the values for $\cos(\theta)$ are only given down to $\cos(\theta)=0.2$, which means θ about 78 degrees. In Figures 2d, e and f, however, values up to $\theta=90$ degrees are given. Where do the data come from?

3.6 In addition, for $\theta=90$ degrees, one has from equation 2 of the manuscript $H_{-+} + H_{--} = 0$ because there, $\cos(\theta)=0$. Apparently, however, $H_{-+} + H_{--} = 2 \times H_{-}$ is true for $\theta=0$.

3.7 The main result of an anisotropic torque relies on the minimum of $H_{--} - H_{-+}$ at small values of $\cos(\theta)$ (Figure 2c). This is in contrast with other publications such as Phys. Rev. Lett. 109, 096602 (2012). The only difference to the PRL paper is the use of SiO₂ instead of Al₂O₃. This issue therefore has to be addressed in the manuscript.

3.8 At my point of view, a detailed structural analysis is also missing (at least in the supplement).

In total, the paper at my point of view does not meet the criteria for Nature Communications and I cannot recommend the paper for publication. After careful rewriting and probably also after some additional experimental evaluation ((Figure 1b, structural analysis etc.), I would recommend transfer to another journal such as Scientific Reports.

Reviewer #2 (Remarks to the Author):

This manuscript “Giant current-induced antidamping orbital torque in ferromagnetic heterostructures” by Chen et al. studied the magnetic field angle and temperature dependence of the torque efficiency by means of magnetization switching measurement in ferromagnetic hetero-structures. The research topic about magnetization control in thin ferromagnetic film with strong perpendicular magnetic anisotropy is very important for spintronics applications. In this paper, Chen et al. reported the giant and anisotropic torque due to new mechanism of “orbital Rashba-Edelstein effect”. For the reason, this work may deserve for publication in Nature Communications. However, the author’s explanation and discussion for experimental results is not enough at following points.

1. Origin of torque is the exchange interaction between magnetization and Δ_L ?
If it is correct, why generated torque become to “anti-damping torque” not field torque?
2. In Fig. 3 c, why is there the different temperature dependence between T_0 and T_2 ? And H_2 in fig. 4d has peak around 120 K. But in the plots of T_2 and T_4 , there is no peak around 120 K. If T^{AD} and H^{en} have the same origin, I think its trend should be similar. If you have some idea, you should add the explanation or comments.
3. Fig. 2f shows the anisotropic efficiency. Could you fit the experimental data by considering the Fermi-contour shift shown in fig.4? I think that the explanation about “anisotropic” property is not enough.
4. In this manuscript, it is not clear why such giant effect was caused by the orbital REE.

Reviewer #3 (Remarks to the Author):

The authors report a mechanism driven by the orbital Rashba-Edelstein effect for highly efficient current-induced magnetization switching in Pt/Co/SiO₂/Pt heterostructures.

The obtained large torque efficiency is due to the intrinsic Co 3d orbital anisotropy and more than an order of magnitude higher than the values previously reported.

I find the results interesting. In particular, large orbital torque reported in this manuscript could shed new light on spintronics but I have several comments:

-The role of the external magnetic field is unclear. In particular, why was no CIMS observed

when $H^{\text{ext}} = 0$?

-It would be helpful to include the definitions of antidamping and field-like torques.

-In Fig. 1a, it would be helpful to clarify whether the current flows parallel or perpendicular to the heterostructures.

-As for the explanation on Figs. 1c and 1d, it would be helpful to clarify how θ changes in the CIMS process (what is the initial value of θ ?).

-While there is some deviation between the data and fit curve in Fig. 2c, they coincide almost completely in Fig. 2d. Why?

-At line 133, how is the formula of the efficiency obtained?

-At line 147, it is written that T^{AD}_4 decreases with decreasing temperature but Fig. 3c does not look so. Also, it is written that H^{an}_2 increases as temperature is decreased but Fig. 3d does not look so.

-At line 182, it would be helpful to add the definition of sp-metals.

Responses to reviewers' comments

Dear reviewers,

We appreciate the helpful comments from you and revised the manuscript significantly to clarify our finding. The detailed responses are given below. Please note that the manuscript title has been revised to “Giant antidamping orbital torque originating from the orbital Rashba-Edelstein effect in ferromagnetic heterostructures”.

Reviewer #1:

The paper discusses Spin-Orbit-Torque switching of a magnetically perpendicular Co film sandwiched between Pt and SiO₂. The authors find low critical current density for switching and discuss the results within a macrospin-model. They evaluate the magnetic anisotropy field and the torques within this model. The two main results are a high torque efficiency and an anisotropic torque. While the paper is well written, the scientific content is not novel enough and the conclusions are not unambiguous enough to justify publication in Nature Communications.

- 1. The authors claim for their critical current density “These j_c values are much smaller than those obtained through the SHE mechanism ...”. It is true, that the j_c values shown in this paper are relatively low. Similar values, however, have been already published (see, e.g., Appl. Phys. Lett. 109, 142405 (2016)). Moreover, the samples discussed in this manuscript cannot be directly compared with those of other publications. A look into ref. 11 of the manuscript (Phys. Rev. Lett. 109, 096602 (2012)) shows, that the samples discussed in the PRL (Pt/Co/Al₂O₃) exhibit a coercive field of more than 150 Oe, while the samples discussed in the present manuscript have only around 25 Oe. That means, that the effective anisotropy, which determines the switching current density is much smaller in the samples discussed in the manuscript than those of, e.g., ref. 11.**

Response:

The critical current density (j_c) is proportional to the intrinsic Gilbert damping (α) and effective magnetic anisotropy energy (K_{eff}), and inversely scales with the torque efficiency (ζ), that is,

$$j_c \propto \alpha K_{\text{eff}} / \zeta \quad (\text{R1})$$

In general, Ta- and W-based heavy-metal (HM)/ferromagnet (FM)/oxide structures have smaller α and K_{eff} than Pt-based ones. The α and K_{eff} of Ta- and W-based structures with perpendicular magnetic anisotropy (PMA) lie in the range of 0.01 – 0.05 and $0.5 - 3 \times 10^6$ erg/cm³, respectively (*Phys. Rev. B* 89, 174416 (2016); *Nat. Mater.* 9, 721 (2010); *Appl. Phys. Lett.* 109, 202404 (2016); *arXiv:1709.07483*; *Appl. Phys. Lett.* 105, 092402 (2014); *Appl. Phys. Lett.* 102, 022407 (2013); *Appl. Phys. Lett.* 105, 052415 (2014); *AIP Advances* 2, 032151 (2012); *Appl. Phys. Lett.* 108, 152405 (2016); *Appl. Phys. Lett.* 109, 142405 (2016); *Appl. Phys. Lett.* 109, 192405 (2016)), while those of Pt-based structures with PMA can reach 0.1~0.3 and $3\sim 10 \times 10^6$ erg/cm³, respectively (*Appl. Phys. Lett.* 102, 082405 (2013); *Appl. Phys. Lett.* 96, 152502 (2010); *Phys. Rev. B* 92, 064426 (2015); *Appl. Phys. Lett.* 104, 092403 (2014); *Appl. Phys. Lett.* 104, 052413 (2014); *Phys. Rev. B* 79, 024423 (2009); *Phys. Rev. B* 95, 094417 (2017); *IEEE Transactions on Magnetics* 44, 2865 (2008)). Consequently, the Ta- and W-based films typically have smaller j_c compared to the Pt-based ones; the j_c of the former can be as low as the order of 10^6 A/cm², whereas that of the latter is rarely reported to be in the same order (usually in the order of 10^7 A/cm²). In this manuscript, we reported j_c values of the order of 10^6 A/cm² in Pt/Co/SiO₂/Pt films, the same order as reported in Ta- and W-based structures.

We agree that our Pt/Co/SiO₂/Pt film has smaller coercivity field than the Pt/Co/AlO_x film in the PRL paper (*Phys. Rev. Lett.* 109, 096602 (2012)). However, our samples have higher PMA than the Pt/Co/AlO_x film. The anisotropy field (H^{an}) and the saturation magnetization (M_s) of the Pt/Co/AlO_x film were 2.8 kOe and 1000 emu/cm³, respectively. Using the formula $K_{\text{eff}} = H^{\text{an}}M_s/2$, the K_{eff} of the Pt/Co/AlO_x film was estimated to be 1.4×10^6 erg/cm³ at room temperature, twice smaller than our case (3.92×10^6 erg/cm³). Even so, we still have lower j_c than the Pt/Co/AlO_x film. The j_c value of the Pt/Co/AlO_x film was reported to be higher than 2.3×10^7 A/cm² with an in-plane external magnetic field (H^{ext}) of 100 Oe. In our Pt/Co/SiO₂/Pt film, the j_c value was found to be 4.9×10^6 A/cm² at $H^{\text{ext}} = 100$ Oe (see Fig. 5 of the revised manuscript). As already discussed in the manuscript, this benefits by the large ζ in the Pt/Co/SiO₂/Pt.

To be clear, one of the important results being reported here is the very large ζ in the Pt/Co/SiO₂/Pt films, and we are not intentional to compare the j_c values with other HM/FM/oxide systems, as also mentioned by the reviewer that “the samples discussed in this manuscript cannot be directly compared with those of other publications”. We fear our plot have misled the reviewer.

For clarification, we have modified the manuscript as follow:

- ✧ The sentence “These j_c values are much smaller than those obtained through the SHE mechanism” in the previous version of the manuscript (Page 5, line 97) has been modified to “These j_c values are much smaller than those previously reported in Pt-based HM/FM/oxide

heterostructures” in the revised manuscript (Page 12, line 243). The words “SHE mechanism” has been deleted, because the mechanisms of the in-plane current-induced torque in HM/FM/oxide structures are still under debate. The words “Pt-based HM/FM/oxide heterostructures” were newly added to clarify the level of our j_c values in Pt-based structures.

✧ The current-induced magnetization switching measurement was moved to Fig. 5 in the revised manuscript just as a supporting experiment that confirms the large ζ .

2. In contrast with this apparent coercive field H_c , the authors evaluate an anisotropy field H^{an} varying between 7 kOe and 10 kOe depending on the angle between magnetization and z-axis. When the magnetization is perpendicular, then H^{an} is 10 kOe, *i.e.* almost three orders of magnitude larger than H_c . In such cases, the magnetization switching is obviously governed by thermal activation and arbitrarily low critical current densities for SOT switching can be achieved if the current flows for a long time. In other words, the values of j_c will depend strongly on the pulse duration as well as on the temperature with j_c going to zero as T approaches the temperature, where H_c becomes zero (around 400K in this case, see Figure 3b of the manuscript). Such thermally activated behaviour was already discussed in the literature for CoFeB/MgO based tunnel junctions (see Timopheev, A.A. et al., Phys. Rev. B 96, 014412 (2017)) and attributed to a crossover from a homogeneous film to a granular type. There, the magnetization dynamics is governed by small activation energies of granules instead of the anisotropy energy of the complete film. Therefore, the main conclusion of the paper, *i.e.* a large value of the torque efficiency is highly questionable.

Response:

In the manuscript we adopted a quasistatic DC current sweeping method (see, e.g., Phys. Rev. Lett. 109, 096602 (2012), Science 336, 555 (2012)) for current-induced magnetization switching (CIMS) measurements instead of using current pulses. To investigate the thermal effects, we studied the dependences of longitudinal resistance R_{yy} on temperature and current magnitude. Fig. R1a shows a top view of a Pt (5)/Co (1)/SiO₂ (1)/Pt (1) (nm) device and the measurement scheme, where V_{yy} is the longitudinal voltage and I is the injected current. The device was placed in a PPMS (physical property measurement system) chamber. During the temperature-dependent measurement, a small I with magnitude of 0.1 mA was used to excite the Hall effect while minimize the Joule heating. In the current-dependent measurement, the applied I started at 0.1 mA with chamber temperature of 300.1 K. Figs. R1b, R1c present temperature- and current-dependent R_{yy} ($R_{yy} = V_{yy}/I$), respectively. R_{yy} shows a linear increase with increasing temperature,

a typical phenomenon for metallic materials. The linear relation between R_{yy} and I^2 shown in the inset of Fig. R1c indicates that the increase of R_{yy} is mainly caused by Joule heating. Comparison of the two figures suggests that a 4-mA current injection leads to a temperature increase of 6 K. Notice that the currents used for torque efficiency evaluation (see Figs. 3a, 3b of the revised manuscript) are 0.9 and 1.5 mA, far below 4 mA. Also, a 4-mA current injection corresponds to a current density of 6×10^6 A/cm², which is sufficiently large to switch the magnetization for all external magnetic field (H^{ext}) values we applied (see Figs. 5a, 5b of the revised manuscript). Therefore, we think that the thermal effects should have little influence on the evaluations of the critical current density (j_c) and torque efficiency, and the low j_c values are the consequence of the very large torque efficiency (up to 2.83, see Fig. 3f of the revised manuscript).

In particular, we like to point out that the magnetization rotation mode is different in the measurements for CIMS and torque efficiency evaluation. In the CIMS measurement, the magnetization switching is achieved by the nucleation and propagation of domain walls rather than by coherent magnetization rotation. If the magnetization switching is achieved by the coherent magnetization rotation mode, the coercivity field (H_c) is expected to be the same to the magnetic anisotropy field (H^{an}). But as shown in Figs. 2a, 2b of the revised manuscript where H^{ext} is perpendicular to the film plane, the magnetization switches sharply and the H_c is significantly smaller than H^{an} , as also noticed by the reviewer. In the torque efficiency evaluation measurement, however, the H^{ext} is applied almost parallel to the y -axis. As shown in Fig. 3a of the revised manuscript, the anomalous Hall effect signal decreases slowly with increasing H^{ext} , strongly evidenced that the magnetization rotates coherently, i.e., no domain walls form during the rotation process. In other words, the torque efficiency evaluation does not rely on the thermal activation process that favor domain wall formation.

Figure R1. **a** Top view and measurement scheme of a Pt(5)/Co(1)/SiO₂(1)/Pt(1) (nm) device. **b** temperature and **c** current (I) dependence of R_{yy} . Inset of **c**: Variation of R_{yy} as a function of I^2 .

Next, we discuss the quality of the Co layer in our Pt/Co/SiO₂/Pt film. In the PRB paper (*Phys. Rev. B* 96, 014412 (2017)), the CoFeB film was found to be magnetically homogeneous when its thickness was above ~ 1.3 nm. When its thickness decreased to below ~ 1.3 nm, multistep-like magnetization switching and superparamagnetic behaviors appeared, and the CoFeB layer became magnetically inhomogeneous. The authors of the PRB paper attributed this phenomenon to a crossover from a continuous film to a granular-type film and the weak coupling between the thermally-unstable granules due to reduced dimension. To investigate the quality of the Co layer of our samples, we performed temperature-dependent measurements of the magnetization and anomalous Hall effect (AHE). The results are shown in Figs. R2a, R2b (also given in Figs. 2a, 2b of the revised manuscript). No superparamagnetic behavior appeared in the magnetization loops even at 400 K, and no multistep-like switching behavior was observed in the AHE curves, indicating that the Co layer is a continuous and thermally-stable film. We also performed element distribution studies using energy-dispersive X-ray spectroscopy. The X-ray imaging of the Co and Pt elements is shown in Fig. R2c (also given in Fig. 1b of the revised manuscript). We found that the Co distributes uniformly and forms a continuous film, supporting our assertion that the Co is a continuous film.

Figure R2. **a** Magnetization loops and **b** anomalous Hall effect measured at 50 K, 300 K, and 400 K in Pt/Co/SiO₂/Pt films. **c** X-ray imaging of Co and Pt.

We would like to give further discussions on the quality of the Co layer from a thermodynamic point of view. It is well known that the morphology and magnetic properties of an ultrathin ferromagnetic layer depend strongly on the underlayers it is deposited on. We take a well-studied MgO/Fe system for example. There, the Fe was grown on top of the MgO. It was found the Fe has a granular-type morphology and exhibits superparamagnetic behavior even at a 10-monolayer thickness (corresponds to 1.43 nm) (*J. Appl. Phys.* 78, 4449 (1995); *J. Appl. Phys.* 84, 1499 (1998); *Surface Science* 498, 193 (2002); *Phys. Rev. B* 52, 12779 (1995); *Phys. Rev. B* 92, 224403 (2015)). When a metallic material such Cr or V was inserted between MgO and Fe, i.e., when the Fe was deposited on top of Cr or V, the Fe was found to form a continuous film and shows ferromagnetic characteristic with strong perpendicular magnetic anisotropy even at a small thickness of 5 monolayer (~0.7 nm) (*Appl. Phys. Lett.* 103, 192401 (2013); *Appl. Phys. Lett.* 102, 122410 (2013); *Appl. Phys. Lett.* 103, 192401 (2013); *Appl. Phys. Lett.* 103, 062402 (2013); *Applied Physics Letters* 105, 122408 (2014)). From the thermodynamic point of view, the granular formation of Fe on MgO originates from the large surface free energy of Fe (2.94 J/m²) (*Jpn. J. Appl. Phys.* 21, 1569 (1982)), compared to MgO (1.1 J/m²) (*Chem. Rev.* 75, 547 (1975)). A large difference in surface free energy prevents the complete wetting of the surfaces when one material is deposited over the other and produces layers with less homogeneous. However, the surface free energies of Cr and V are 2.05 and 2.87 J/m² (*Jpn. J. Appl. Phys.* 21, 1569 (1982)), respectively, close to that of Fe. Consequently, the Fe tends to form a continuous film and has enhanced magnetic anisotropy when grown on Cr or V.

In the PRB paper, the CoFeB film was deposited on MgO. Although there is no data reported for the surface free energy of CoFeB, we deduce that it should be significantly different from that of MgO, because not only the PRB paper but also other published works (*Appl. Phys. Lett.* 89, 163119 (2006); *J. Appl. Phys.* 107, 113718 (2010); *IEEE Transactions on Magnetics* 46, 2116 (2010); *IEEE Transactions on Magnetics* 50, 1401404 (2014)) have demonstrated that ultrathin CoFeB films grown on MgO usually exhibit a granular-like and superparamagnetic behavior when its thickness is below ~ 1.3 nm. In our Pt/Co/SiO₂/Pt film, the Co was deposited on the metallic Pt. The surface free energy of Co and Pt are 2.71 and 2.69 J/m² (*Jpn. J. Appl. Phys.* 21, 1569 (1982)), very close to each other. Therefore, it is expected that the Co can form a continuous film on the Pt, verified by the temperature-dependent magnetic properties and element distribution analyses in Fig. R2.

In summary, the analyses on the temperature-dependent magnetic properties, the X-ray element imaging, and the thermodynamics reveal that the Co layer is a continuous film and thermally-stable. The torque efficiency evaluation does not depend on the thermally-activated domain wall formation mode but on the coherent magnetization rotation process. Therefore, the result of the large torque efficiency in our work is believed to be reliable.

we have revised several sentences and added more discussions in the manuscript to make these points more clearly:

- ✧ Page 8, line 156: The sentence “Contrast to the sharp switching in Fig. 2b, the slow decrease of R_{AHE} with increasing H^{ext} observed in Figs. 3a, 3b is due to the coherent rotation of the Co magnetization towards the hard magnetization axis” is added.
- ✧ Page 11, line 233: The sentence “In order to confirm the large T^{AD} and ξ^{AD} , we performed CIMS experiments at 300 K, where the magnetization direction was monitored by measuring R_{AHE} using a quasistatic DC current sweeping method with a ramp rate of 0.5 mA/s” is added.
- ✧ Supplementary information: Section S5 is added to discuss the thermal effect.
- ✧ Page 12, line 246 to 254: The sentences “Note here that the increase in device temperature due to current-induced Joule heating ... and thus did not influence upon the estimations of T^{AD} and ξ^{AD} ” are added.
- ✧ Figs. 1a, 1b are added to investigate the crystal structures and elemental distributions in the Pt/Co/SiO₂/Pt films. The X-ray photoelectron spectroscopy spectrum presented in Fig. 4b in the previous version of the manuscript is moved forward to Fig. 1c in the revised version.
- ✧ Figs. 2a, 2b are added to investigate the temperature-dependent magnetic properties.

- ✧ Page 6, line 111 to 126: The sentences “Ultrathin metallic FM layers with thickness less than 1.5 nm...we believe that the granular effect [29,30] has little influence on the evaluations of ζ and MCA in our work (see below)” are added.
- ✧ Page 7, line 129 to 132: The sentence “We notice that there is a difference in H_c between the MH and R_{AHE} loops, which we ascribe to the different sample size for MH and AHE measurements, the microfabrication process, and the measurement technique used” is added.

3. While the discussion of the critical current density is, therefore, somehow meaningless, the evaluation of the anisotropy and the torques is within a macrospin model and has scientific value. In particular the finding of an anisotropic torque seems interesting. There are, however, some questions arising:

3.1 In Figure 1b, there is a substantial difference between the M(H) loop and the loop from the anomalous Hall effect. This cannot be true if the same sample was used for the measurement. I suspect, that the processing of the M(H) data (cancelling of para- or diamagnetic background) is the reason.

Response:

Actually, the samples used for $M-H$ and anomalous Hall effect (AHE) measurements were different (see the Method section). The diamagnetic background has also been subtracted. In our opinion, the difference between the $M-H$ and AHE loops is caused by the sample size used for measurement, the microfabrication process, and the measurement technique used. The sample used for $M-H$ measurement has an area of $3 \times 4 \text{ mm}^2$ and was measured using a vibrating sample magnetometer. The sample for AHE measurement was patterned into Hall devices with channel width of $10 \text{ }\mu\text{m}$ and length of $80 \text{ }\mu\text{m}$ using ultraviolet lithography and ion beam etching, and measured using an electrical method. It is known that the vibrating sample magnetometer detects the whole magnetic signals from both the sample and substrate while the AHE is only sensitive to the out-of-plane component of the magnetization. Moreover, during the Hall device fabrication process, the samples are required to heat up to 100°C to solidify the photoresist and remove the deionized water on the sample surface, then followed by an argon ion etching. However, no such treatments were applied to the samples used for $M-H$ measurement.

We have added and revised several sentences in the manuscript to make this point more clearly:

- ✧ Page 7, line 129 to 132: The sentence “We notice that there is a difference in H_c between the MH and R_{AHE} loops, which we ascribe to the different sample size for MH and AHE measurements, the microfabrication process, and the measurement technique used” is added.
- ✧ Page 20, line 394: The sentence “During the microfabrication process, the films were heated to 100 °C to solidify the photoresist and remove the deionized water at surface” is added.
- ✧ Page 20, line 411: The sentence “Note here that the sample area in electrical and magnetic measurements is different” is added.

3.2 The temperature, at which the experiments have been done must be given for all results, because both the coercive fields as well as the critical switching currents strongly depend on temperature.

Response:

We have indicated the temperature at which the experiments have been done in the revised manuscript.

3.3 To understand the data, it is necessary to also give the shape of the current pulses that have been driven through the sample.

Response:

In this study, we adopted a quasistatic DC current sweeping method instead of using current pulses. We used a Keithley 2400 sourcemeter for DC current sweeping and a Keithley 2182A nanovoltmeter for voltage acquisition. The current sweeping rate was 0.5 mA/s.

3.4 The control samples are not really of value for the paper, because the discussion is not deep enough. Do the authors really suggest, that a sample such as Pt/Co/Pt has no contribution of the interfaces that favour a PMA at small Co thickness? This would be in contrast to all experimental observations (see, e.g., J. Magnet. Mater. 93, 194 (1991)).

Response:

The controlled samples Pt/Co/Pt and SiO₂/Co/SiO₂ were used to illustrate the critical role of asymmetric orbital hybridizations at the Pt/Co and Co/SiO₂ interfaces in achieving large orbital splitting and, thus, large current-induced orbital torque. In the Pt/Co/Pt film, the Pt/Co and Co/Pt interfaces have symmetrical 3d-5d orbital hybridization but opposite direction of interface crystal

electric field \mathbf{E}_{CEF} (please refer to the inset of Fig. 6b in the revised manuscript). The total \mathbf{E}_{CEF} therefore tends to cancel out and thereby reduces the Co 3d orbital splitting, as the orbital splitting is governed by \mathbf{E}_{CEF} (see Fig. 6d in the revised manuscript). Although the Pt/Co interface favor a perpendicular magnetic anisotropy (PMA), the reduced orbital splitting cannot overcome the demagnetization field of the 1-nm-thick Co layer, resulting in the in-plane magnetic anisotropy in the Pt/Co/Pt film. This is also the case in the SiO₂/Co/SiO₂ film (please refer to Fig. 6c in the revised manuscript). In the Pt/Co/SiO₂/Pt film, by contrast, the Pt/Co and Co/SiO₂ interfaces have asymmetric orbital hybridizations and same signs of \mathbf{E}_{CEF} . As a result, the total \mathbf{E}_{CEF} increases in the Pt/Co/SiO₂/Pt film (see Fig. 6a in the revised manuscript) and induces a large Co 3d orbital splitting, which not only stabilizes the magnetization at the normal direction (i.e., the establishment of PMA) but also gives rise to a large orbital polarization in the presence of an in-plane current (i.e., the orbital Rashba-Edelstein effect, please refer to Fig. 6e in the revised manuscript). The generated orbital polarization then couples with the Co magnetization via exchange interaction and exerts a large orbital torque on the Co magnetization.

In the revised manuscript, we have provided an improved discussion on the controlled sample:

- ✧ Page 6, line 102 to 105: The sentence “As will be shown in this study, the combined effect of the Pt 5d-Co 3d-O 2p orbital hybridization can produce a large orbital splitting in the Co layer, which not only establishes the PMA of the Pt/Co/SiO₂/Pt films but also induces a very large torque on the Co magnetization in the presence of an in-plane current” is added.
- ✧ Page 15, line 312 to 333: The sentences “We next discuss the giant effects of the orbital torque ... and is enhanced due to the asymmetric orbital hybridizations at the Pt/Co and Co/SiO₂ interfaces” are added.
- ✧ The plot of Fig. 6 has been improved.

3.5 In Figure 2a, the values for cos(theta) are only given down to cos(theta)=0.2, which means theta about 78 degrees. In Figures 2d, e and f, however, values up to theta=90 degrees are given. Where do the data come from?

Response:

Values with $\theta > 78^\circ$ were given based on the torque symmetry. When considering high-order terms in the current-induced torque and space inversion symmetries (*Nature Nanotech.* 8, 587 (2013)), the antidamping torque \mathbf{T}^{AD} has the form $\mathbf{T}^{\text{AD}} = \mathbf{m} \times (\boldsymbol{\sigma} \times \mathbf{m}) T_0^{\text{AD}} + (\mathbf{z} \times \mathbf{m})(\mathbf{m} \cdot \mathbf{y})[T_2^{\text{AD}} + T_4^{\text{AD}}(\mathbf{z} \times \mathbf{m})^2] = (T_0^{\text{AD}} + T_2^{\text{AD}} \sin^2\theta + T_4^{\text{AD}} \sin^4\theta)\mathbf{x} = T^{\text{AD}}\mathbf{x}$, with current applied along the \mathbf{y} -axis and magnetization unit vector \mathbf{m} rotating in the yz -plane (please refer to Section S1 in Supplementary

Information). Here, $T^{\text{AD}} = T_0^{\text{AD}} + T_2^{\text{AD}} \sin^2\theta + T_4^{\text{AD}} \sin^4\theta$ describes the magnitude of \mathbf{T}^{AD} , $\boldsymbol{\sigma}$ is the unit vector of the current-induced orbital polarization, and T_0^{AD} , T_2^{AD} , and T_4^{AD} are the zeroth-, second-, and fourth-order terms in T^{AD} . So, once having obtained the coefficients T_0^{AD} , T_2^{AD} , and T_4^{AD} , we can plot the angular distribution of T^{AD} according to its space symmetry, closely resembling the well-known situation of the uniaxial magnetic anisotropy energy $K_u = K_1 \sin^2\theta + K_2 \sin^4\theta + K_3 \sin^6\theta$, where K_1 , K_2 , and K_3 are the anisotropy constants.

We have revised several sentences in the manuscript to make this point more clearly:

- ✧ Page 9, line 166 to 169: The sentence ‘‘When considering the space symmetry of torques [23], \mathbf{T}^{DL} takes the form $\mathbf{T}^{\text{AD}} = \mathbf{m} \times (\boldsymbol{\sigma} \times \mathbf{m}) T_0^{\text{AD}} + (\mathbf{z} \times \mathbf{m})(\mathbf{m} \cdot \mathbf{y})[T_2^{\text{AD}} + T_4^{\text{AD}} (\mathbf{z} \times \mathbf{m})^2]$, where T_0^{AD} , T_2^{AD} , and T_4^{AD} correspond to the zeroth-, second-, and fourth-order terms in T^{AD} , respectively’’ is added.
- ✧ Page 10, line 192: The sentence ‘‘Having obtained the coefficients T_n^{AD} and H_n^{an} ($n = 0, 2,$ and 4), we can investigate the detailed angular distributions of T^{AD} and H^{an} based on their space symmetries and, more importantly, the possible interrelation between the two quantities’’ is revised.

3.6 In addition, for theta=90 degrees, one has from equation 2 of the manuscript $\mathbf{H}_{-+} + \mathbf{H}_{--} = \mathbf{0}$ because there, $\cos(\text{theta})=0$. Apparently, however, $\mathbf{H}_{-+} + \mathbf{H}_{--} = 2 \times \mathbf{H}_{\text{an}}$ is true for theta=0.

Response:

We are unclear if you meant ‘‘ $H_{-+} + H_{--} = 2 \times H_{\text{an}}$ is true for theta=90 degree’’? If so, we have the following explanations for this comment. In the Hall measurements shown in Figs. 3a, 3b of the revised manuscript, the external magnetic field (\mathbf{H}^{ext}) was slightly tilted from the xy -plane with a small angle $\beta = 3^\circ$. This means the allowed θ for magnetization rotation is in the range of $0 - 87^\circ$ as shown in Fig. R3a, where θ is the angle between the magnetization and \mathbf{z} . For the case where the magnetization can rotate to $\theta = 90^\circ$, it requires $\beta = 0$, i.e., \mathbf{H}^{ext} is parallel to the y -axis as shown in Fig. R3b. Substituting $\beta = 0$ into the equation

$$H_-^{\text{ext}}(\theta) + H_+^{\text{ext}}(\theta) = 2(H_0^{\text{an}} + H_2^{\text{an}} \sin^2\theta + H_4^{\text{an}} \sin^4\theta) \cos\theta \sin\theta / \cos(\theta + \beta),$$

we have

$$H_-^{\text{ext}}(\theta) + H_+^{\text{ext}}(\theta) = 2(H_0^{\text{an}} + H_2^{\text{an}} \sin^2\theta + H_4^{\text{an}} \sin^4\theta) \sin\theta.$$

Now, we have $H_-^{\text{ext}}(\theta) + H_+^{\text{ext}}(\theta) = 2(H_0^{\text{an}} + H_2^{\text{an}} + H_4^{\text{an}}) = 2H^{\text{an}}$ for $\theta = 90^\circ$.

Figure R3. Sketches of the allowed θ range for magnetization rotation when **a**, $\beta = 3^\circ$ and **b**, $\beta = 0$.

3.7 The main result of an anisotropic torque relies on the minimum of $H_{-} - H_{+}$ at small values of $\cos(\theta)$ (Figure 2c). This is in contrast with other publications such as Phys. Rev. Lett. 109, 096602 (2012). The only difference to the PRL paper is the use of SiO_2 instead of Al_2O_3 . This issue therefore has to be addressed in the manuscript.

Response:

In our opinion, this diversity comes from the difference in the electronic structures of films. We take the PRL paper (*Phys. Rev. Lett.* 109, 096602 (2012)) for comparison. In the Pt/Co/ AlO_x , a metallic Al capping layer was first deposited on Co and subsequently oxidized into AlO_x by exposure to the atmosphere. In our Pt/Co/ SiO_2 /Pt, the SiO_2 layer was directly deposited from a SiO_2 target using RF sputtering. After preparation, the Pt/Co/ AlO_x was annealed at 350 °C for 1 hour while in our study, the Pt/Co/ SiO_2 /Pt was annealed at 200 °C for 30 minutes. With regard to the interface electronic structures, the X-ray photoelectron spectroscopy analyses indicates the presence of Co 3d-O 2p hybridization at the Co/ SiO_2 interface (see Fig. 1c of the revised manuscript). Although there is no such electronic structure characterization in the Pt/Co/ AlO_x , we speculate that it should be very different from the case in our Pt/Co/ SiO_2 /Pt, because the orbital hybridization at the FM/oxide interface is sensitive to the film preparation condition, the oxide layer used, and the annealing treatment (*Rev. Mod. Phys.* 89, 025008 (2017)). Additionally, the Pt and Co layer thicknesses in the Pt/Co/ AlO_x are 2 and 0.6 nm, respectively, while in our Pt/Co/ SiO_2 /Pt, they are 5 and 1 nm, respectively. The difference in the layer thickness can also affect the electronic structures such as the orbital hybridization at the Pt/Co and Co/(SiO_2 or AlO_x) interfaces.

Figure R4. Curve fitting without and with high-order terms in **a** H^{an} and **b** T^{AD} .

Our speculation is supported by the difference in the magnetic properties between the Pt/Co/SiO₂/Pt and Pt/Co/AlO_x. In the PRL paper, the authors showed that their data can be well fitted within a macrospin model where only the isotropic term H_0^{an} was considered (see Fig. 2d in the PRL paper). Since the interface orbital hybridization governs the magnetic properties, this indicates that the high-order effect of the orbital hybridization on the magnetic anisotropy is weak in the Pt/Co/AlO_x. Correlatively, the authors of the PRL paper found that the current-induced torque can also be well understood when considering only the isotropic terms T_0^{AD} . However, we found in our study that the high-order uniaxial terms in H^{an} are required to account for our observations. Without considering high-order uniaxial terms in H^{an} , i.e., $H_2^{\text{an}} = H_4^{\text{an}} = 0$, the Eq. 3 of the revised manuscript, i.e.,

$$H_-^{\text{ext}}(\theta) + H_+^{\text{ext}}(\theta) = 2(H_0^{\text{an}} + H_2^{\text{an}} \sin^2 \theta + H_4^{\text{an}} \sin^4 \theta) \cos \theta \sin \theta / \cos(\theta + \beta)$$

simplifies to

$$H_-^{\text{ext}}(\theta) + H_+^{\text{ext}}(\theta) = 2H_0^{\text{an}} \cos \theta \sin \theta / \cos(\theta + \beta),$$

which corresponds to the case in the PRL paper where only the H_0^{an} was considered. As shown in Fig. R4a, the fitting curve without high-order uniaxial terms deviates significantly from the data points. This is also the case in the T^{AD} fitting shown in Fig. R4b when not considering the high-order terms T_2^{AD} and T_4^{AD} in T^{AD} , indicating that the high-order terms of the orbital hybridization have strong effects on the T^{AD} in our Pt/Co/SiO₂/Pt.

We have added more discussions in the revised manuscript and supplementary information to make this point more clearly:

- ✧ Page 9, line 182: The sentence “The result of the anisotropic T^{AD} reported here is in contrast to previous studies where T^{AD} was demonstrated to be angular-independent [8,9]. We

ascribed such diversity to the difference in the electronic structures of films (see Supplementary Section S7)” is added.

- ✧ Supplementary information: Section S7 is added to give comments on previous experiments where the current-induced antidamping torque was found to be angular-independent.

3.8 At my point of view, a detailed structural analysis is also missing (at least in the supplement).

Response:

Following your recommendations, we have performed high-resolution transmission electron microscopy study and X-ray element mapping to investigate the crystal structure and element distribution of our Pt/Co/SiO₂/Pt films, respectively. The results are presented in Figs. 1a, 1b in the revised manuscript. Also, the X-ray photoelectron spectroscopy spectrum given in Fig. 4b in the previous version of the manuscript was moved forward to Fig. 1c in the revised one. For detailed discussions, please refer to the “Crystal and interface electronic structures” section in the revised manuscript.

Reviewer #2

This manuscript “Giant current-induced antidamping orbital torque in ferromagnetic heterostructures” by Chen et al. studied the magnetic field angle and temperature dependence of the torque efficiency by means of magnetization switching measurement in ferromagnetic hetero-structures. The research topic about magnetization control in thin ferromagnetic film with strong perpendicular magnetic anisotropy is very important for spintronics applications. In this paper, Chen et al. reported the giant and anisotropic torque due to new mechanism of “orbital Rashba-Edelstein effect”. For the reason, this work may deserve for publication in Nature Communications. However, the author’s explanation and discussion for experimental results is not enough at following points.

We appreciate your positive review that our manuscript may deserve for publication in Nature Communications! The answers to the questions you raised are detailed here.

1 Origin of torque is the exchange interaction between magnetization and ΔL ? If it is correct, why generated torque become to “anti-damping torque” not field torque?

Response:

Yes. We believe in our study that the antidamping torque originates from the exchange interaction between the magnetization and current-induced orbital polarization (ΔL). In generally, the in-plane current-induced torque in heavy-metal/ferromagnet/oxide heterostructures, regardless of detailed mechanism, contains two orthogonal components: an antidamping-like component T^{AD} and a field-like component T^{FL} . The magnitude of the two torque components is strongly material- and interface-dependent. Some typical examples are as follows. Miron et al. (*Nature Mater.* 9, 230 (2010)) reported a giant T^{FL} with magnitude of 1 T per 10^8 A/cm² in a Pt/Co/ AlO_x structure, where the Pt thickness was 3 nm and the AlO_x was prepared using an oxygen radiofrequency plasma. However, Liu et al. (*Phys. Rev. Lett.* 109, 096602 (2012)) reported a negligibly small T^{FL} in the same structures, where the Pt thickness was 2 nm and the AlO_x was prepared by natural oxidation of a metallic Al in the atmosphere. In Ta/CoFeB/MgO films, Kim et al. (*Nature Mater.* 12, 240 (2013)) found that T^{AD} and T^{FL} are both large, and the torque magnitude and sign are strongly dependent on the Ta and CoFeB thickness. Ou et al. (*Phys. Rev. B* 94, 140414 (2016)) also reported appreciable T^{AD} and T^{FL} in (Ta, Pt, or W)/CoFeB/MgO films, whose magnitude can be tuned by interface engineering. In this study, we reported a sizable T^{AD} but undetectable T^{FL} . The absence of T^{FL} indicated that the extrinsic angular momentum relaxation mechanism, which mostly generates T^{FL}

according to a drift-diffusion model (*Phys. Rev. B* 87, 174411 (2013); *Phys. Rev. Lett.* 108, 117201 (2012)), is ineffective in our Pt/Co/SiO₂/Pt films.

- 2 In Fig. 3c, why is there the different temperature dependence between T0 and T2? And H2 in fig. 4d has peak around 120 K. But in the plots of T2 and T4, there is no peak around 120 K. If T^{AD} and H^{an} have the same origin, I think its trend should be similar. If you have some idea, you should add the explanation or comments.**

Response:

In this study, we ascribed the current-induced orbital torque anisotropy to the Co 3d orbital splitting based on the phenomenological analysis that confirms the close relationship between the torque anisotropy and magnetic anisotropy. Because the orbital splitting is determined by the crystal electric field (please refer to Fig. 6 and the corresponding discussions in the revised manuscript), we think that the different temperature dependence between T_0^{AD} , T_2^{AD} , and T_4^{AD} may have originated from the different temperature dependence between lowest- and high-order crystal electric field terms. To our knowledge, the temperature dependence of high-order terms of the current-induced torque in HM/FM/oxide systems has not been studied yet. Therefore, we call for further experimental and theoretical researches on this topic to reveal its exact origins.

We have improved the relevant discussion in the revised manuscript to make this point more clearly:

- ◇ Page 17, line 353: The sentences “The temperature dependence of the T^{AD} and K_u can be interpreted as a phonon-mediated electron hopping effect ... calculations based on a realistic electronic structure are needed” are added.

- 3 Fig. 2f shows the anisotropic efficiency. Could you fit the experimental data by considering the Fermi-contour shift shown in fig.4? I think that the explanation about “anisotropic” property is not enough.**

Response:

In the manuscript, the Fermi contours were drawn as circles for the sake of simplicity. For real electronic structures, however, the orbital textures at the Fermi surfaces can be very complicated and far beyond perfect circles, especially in 3d orbitals with multiple symmetries (*Nature Commun.* 5, 3414 (2014); *Phys. Rev. B* 90, 205423 (2014); *Phys. Rev. B* 85, 195401 (2012)). Moreover, the exchange interaction can further distort the Fermi contours in our magnetized Pt/Co/SiO₂/Pt films (*Phys. Rev.*

B 93, 125409 (2016); Phys. Rev. X 6, 041048 (2016)), making the orbital textures more complicated. To fit the experimental data using Fermi-contour shift, one has to calculate the current-induced orbital torque based on a realistic description of the electronic structure, which is beyond the scope of our ability.

To make the anisotropic properties more clearly, we have improved the relevant discussions in the manuscript:

- ✧ **Fig. 6f** is added to qualitatively illustrate the generation of the anisotropic orbital torque. For detailed discussions, please refer to the “Mechanisms of the torque anisotropy and its strong temperature dependence” section in the revised manuscript. There, we interpreted the anisotropic orbital torque as a result of the magnetization direction-dependent Fermi contour distortion effect.

4 In this manuscript, it is not clear why such giant effect was caused by the orbital REE.

Response:

In this study, we observed a giant, anisotropic, and strongly temperature-dependent current-induced antidamping torque in Pt/Co/SiO₂/Pt heterostructures with strong perpendicular magnetic anisotropy. These phenomena first ruled out the spin Hall effect as the dominant mechanism of the current-induced antidamping torque. This is because: (1) the spin Hall angle of Pt rarely exceeds 0.3 according to previous reports (*for a review on the spin Hall angle, see Rev. Mod. Phys. 87, 1213 (2015)*), far below the torque efficiency (up to 2.83) observed in our Pt/Co/SiO₂/Pt; (2) theoretical calculations predicted the current-induced torque due to spin Hall effect not to have any angular dependence (*Phys. Rev. B 87, 174411 (2013)*); (3) previous experiments have demonstrated that the spin Hall effect of Pt is almost temperature-independent (*Phys. Rev. Lett. 99, 226604 (2007); Appl. Phys. Lett. 105, 152412 (2014)*). Moreover, we found that the field-like torque was negligibly small in our Pt/Co/SiO₂/Pt heterostructures, indicating that the basic spin Rashba effect, in the context of which the field-like torque would be sizable (*Phys. Rev. B 87, 174411 (2013); Appl. Phys. Lett. 102, 252403 (2013)*), cannot be accounted to our observation.

Electrons have orbital moments in addition to spin moments. Further, in contrast to the isotropic and weakly temperature-dependence spin moments, the orbital moment is naturally anisotropic and strongly temperature-dependent. More importantly, we observed a strong correlation between the current-induced antidamping torque and magnetic anisotropy (please refer to **Fig. 3e** and **Fig. 4c** in the revised manuscript), suggesting that the current-induced antidamping torque may have the

same origin as the magnetic anisotropy, i.e., it may also stem from the orbital anisotropy. Therefore, we proposed an orbital Rashba-Edelstein effect (please refer to Fig. 6 in the revised manuscript), within which we found that the giant, anisotropic, and strongly temperature-dependent current-induced antidamping torque can be qualitatively understood.

According to this model, the key element to achieve high-efficiency torque is a large orbital splitting. In the Pt/Co/SiO₂/Pt, asymmetric orbital hybridization at the Pt/Co and Co/SiO₂ interfaces can cause a large orbital splitting to overcome the demagnetization field and stabilize the magnetization at the normal direction, which is supported by the presence of the perpendicular magnetic anisotropy. The large orbital splitting in the Pt/Co/SiO₂/Pt is further evidenced by two controlled samples Pt/Co/Pt and SiO₂/Co/SiO₂, who exhibit in-plane magnetic anisotropy because of reduced orbital splitting caused by symmetrical interface orbital hybridization (Figs. 6b, 6c in the revised manuscript). Then, when a current is injected along the film plane of the Pt/Co/SiO₂/Pt, the large orbital splitting produces a sizable orbital polarization (i.e., the orbital Rashba-Edelstein effect), which can couple with the magnetization via exchange interaction and gives rise to the giant orbital torque.

We have revised the manuscript to make this point more clearly:

- ✧ Abstract: The sentences “The key element is a pronounced Co 3*d* orbital splitting due to asymmetric orbital hybridization at the Pt/Co and Co/SiO₂ interfaces, which not only stabilizes the PMA but also produces a large orbital torque upon the Co magnetization with current injection” is added.
- ✧ Page 6, line 102: The sentence “As will be shown in this study, the combined effect of the Pt 5*d*-Co 3*d*-O 2*p* orbital hybridization can produce a large orbital splitting in the Co layer, which not only establishes the PMA of the Pt/Co/SiO₂/Pt films but also induces a very large torque on the Co magnetization in the presence of an in-plane current” is added.
- ✧ Page 11, line 218: The sentence “the degree of anisotropy of T_2^{AD} and K_u , defined as $|T_2^{\text{AD}} + T_4^{\text{AD}}|$ [25] and K_{eff} , respectively, was found to possess similar temperature dependence (Fig. 4c). Overall, they increase with decreasing temperature, suggesting that the anisotropic T^{AD} may have the same origin as K_u ” is added.
- ✧ The whole Discussion section of the revised manuscript have been revised substantially. The discussions on the controlled samples have been improved to elucidate the significance of the asymmetric orbital hybridization in the orbital splitting and thus, the giant orbital torque. The discussions on the temperature- and angle-dependent antidamping orbital torque have also been improved.

Reviewer #3

The authors report a mechanism driven by the orbital Rashba-Edelstein effect for highly efficient current-induced magnetization switching in Pt/Co/SiO₂/Pt heterostructures. The obtained large torque efficiency is due to the intrinsic Co 3d orbital anisotropy and more than an order of magnitude higher than the values previously reported.

I find the results interesting. In particular, large orbital torque reported in this manuscript could shed new light on spintronics but I have several comments:

We appreciate your positive review on our manuscript! The answers to the questions you raised are detailed here.

1 The role of the external magnetic field is unclear. In particular, why was no CIMS observed when $H^{\text{ext}} = 0$?

Response:

In heavy-metal (HM)/ferromagnet (FM)/oxide heterostructures with uniaxial perpendicular magnetic anisotropy (PMA), an in-plane current injection can induce an angular momentum polarization (here is the orbital polarization), which exerts an antidamping torque upon the FM magnetization via exchange interaction. The current-induced antidamping torque, in this case, lies in the xy -plane with direction perpendicular to the current path, according to its symmetry $\mathbf{t}^{\text{AD}} = \mathbf{m} \times (\boldsymbol{\sigma} \times \mathbf{m})$. Here, \mathbf{t}^{AD} , \mathbf{m} , and $\boldsymbol{\sigma}$ are the unit vectors of the antidamping torque, the magnetization, and the orbital polarization, respectively. As the injected current increases, the in-plane torque becomes strong enough to bring the FM magnetization in-plane, where it is metastable. If the charge current is now switched off, the FM magnetization, from this metastable state, will randomly go to an ‘up’ or ‘down’ direction, i.e., it cannot switch deterministically, because the time reversal symmetry is still reversed in the xy -plane (*Nature Nanotechnol.* 9, 548 (2014)). To break the in-plane time reversal symmetry, the most widely adopted method is to apply a small external magnetic field (H^{ext}) parallel or antiparallel to the current direction (see, e.g., *Nature* 476, 189 (2011); *Science* 336, 555 (2012); *Phys. Rev. Lett.* 109, 096602 (2012); *Appl. Phys. Lett.* 104, 082407 (2014); *Phys. Rev. Lett.* 119, 077702 (2017); *Phys. Rev. Applied* 7, 014004 (2017)). In this case, deterministic CIMS can be achieved.

The application of extra H^{ext} will make the device design complicated. Therefore, plenty of efforts has been devoted to the realization of deterministic CIMS without extra H^{ext} . At present, a wedge-shape oxide layer, an antiferromagnet layer, or a ferroelectric substrate has been used to

replace the role of the H^{ext} (*Nature Nanotechnol.* 9, 548 (2014); *Nature Mater.* 15, 535 (2016); *Nature Nanotechnol.* 11, 878 (2016); *Nature Commun.* 7, 10854 (2016); *Nature Mater.* 16, 712 (2017)). Deterministic CIMS was also realized in structures with interlayer exchange coupling or tilted magnetic anisotropy (*Nature Nanotechnol.* 11, 758 (2016); *Proc Nat Acad Sci.* 112, 10310 (2015)). In principle, these methods can also be applied in our study to induce deterministic CIMS.

We have improved the manuscript to make this point more clearly:

- ✧ Page 11, line 227: The sentence “The application of the small H^{ext} was to break the time reversal symmetry in the xy -plane so that deterministic CIMS can be achieved” is added.

2 It would be helpful to include the definitions of antidamping and field-like torques.

Response:

In generally, the in-plane current-induced torque in HM/FM/oxide structures, regardless of detailed mechanism, can separate into two orthogonal components. One is an even function of \mathbf{m} expressed as $\mathbf{T}^{\text{AD}} = T^{\text{AD}}\mathbf{m} \times (\boldsymbol{\sigma} \times \mathbf{m})$ and is called the antidamping torque, because it can compensate the intrinsic Gilbert magnetic damping and induces magnetization switching. The other one is an odd function of \mathbf{m} expressed as $\mathbf{T}^{\text{FL}} = T^{\text{FL}}(\mathbf{m} \times \boldsymbol{\sigma})$ and is called the field-like torque, as it can induce magnetization precession like a magnetic field does. Here, \mathbf{m} is the magnetization unit vector, $\boldsymbol{\sigma}$ is the orbital polarization unit vector, and T^{AD} and T^{FL} describe the magnitudes of \mathbf{T}^{AD} and \mathbf{T}^{FL} , respectively. When considering high-order effects and space inversion symmetry (*Nature Nanotech.* 8, 587 (2013)), the form of \mathbf{T}^{AD} is expanded to $\mathbf{T}^{\text{AD}} = \mathbf{m} \times (\boldsymbol{\sigma} \times \mathbf{m}) T_0^{\text{AD}} + (\mathbf{z} \times \mathbf{m})(\mathbf{m} \cdot \mathbf{y})[T_2^{\text{AD}} + T_4^{\text{AD}}(\mathbf{z} \times \mathbf{m})^2] = (T_0^{\text{AD}} + T_2^{\text{AD}} \sin^2\theta + T_4^{\text{AD}} \sin^4\theta)\mathbf{x} = T^{\text{AD}}\mathbf{x}$, when the current is applied along the \mathbf{y} axis and \mathbf{m} rotates in the yz -plane. Here, T_0^{AD} , T_2^{AD} , and T_4^{AD} is the zeroth-, second-, and fourth-order terms of T^{AD} and $T^{\text{AD}} = T_0^{\text{AD}} + T_2^{\text{AD}} \sin^2\theta + T_4^{\text{AD}} \sin^4\theta$.

We have included the definitions of \mathbf{T}^{AD} and \mathbf{T}^{FL} in the revised manuscript:

- ✧ Page 7, line 134 to 147: The sentences “When a charge current is injected along the film plan of HM/FM/oxide heterostructures ... Here, \mathbf{m} is the magnetization unit vector, $\boldsymbol{\sigma}$ is the current-induced angular momentum polarization and T^{AD} and T^{FL} describe the magnitudes of \mathbf{T}^{AD} and \mathbf{T}^{FL} , respectively” are added.
- ✧ Page 9, line 166: The sentences “When considering the space symmetry of torques [23], \mathbf{T}^{DL} takes the form $\mathbf{T}^{\text{AD}} = \mathbf{m} \times (\boldsymbol{\sigma} \times \mathbf{m}) T_0^{\text{AD}} + (\mathbf{z} \times \mathbf{m})(\mathbf{m} \cdot \mathbf{y})[T_2^{\text{AD}} + T_4^{\text{AD}}(\mathbf{z} \times \mathbf{m})^2]$, where T_0^{AD} , T_2^{AD} , and T_4^{AD} correspond to the zeroth-, second-, and fourth-order terms in T^{AD} , respectively. It is straightforward to have $T^{\text{AD}} = T_0^{\text{AD}} + T_2^{\text{AD}} \sin^2\theta + T_4^{\text{AD}} \sin^4\theta$ ” are added.

3 In Fig.1a, it would be helpful to clarify whether the current flows parallel or perpendicular to the heterostructures.

Response:

In this study, the charge current is injected along the y -axis, i.e., parallel to the heterostructures. We have improved relevant figures, e.g., the insets of Figs. 3a, 3b in the revised manuscript, to clarify the current flow direction.

4 As for the explanation on Figs. 1c and 1d, it would be helpful to clarify how θ changes in the CIMS process (what is the initial value of θ ?).

Response:

We performed the following measurement to investigate the initial value of θ in the current-induced magnetization switching (CIMS) process. Here, θ is the angle between the magnetization and the film normal direction. Fig. R5a shows the normalized anomalous Hall effect (i.e., $\cos \theta$) as a function of the external magnetic field (H^{ext}), measured in a Pt/Co/SiO₂/Pt device with $\beta = 3^\circ$ and temperature of 300 K (please refer to the inset of Fig. 3a of the revised manuscript). The measurement current is 0.1 mA, with direction along y . Fig. R5b presents a $\cos \theta$ vs. H^{ext} curve extracted from the first quadrant of Fig. R5a. The curve shows slow decrease with increasing H^{ext} , indicating that the magnetization rotates coherently, so we can use it to estimate the θ value through an inverse trigonometric transformation. The transformation result is shown in Fig. R5c. The θ value increases with increasing H^{ext} . We find that the application of $H^{\text{ext}} = 25, 50, 100, 150, 200, 250, 300, 350,$ and 400 Oe tilts the magnetization from film normal by approximately $0.5^\circ, 0.7^\circ, 1^\circ, 1.3^\circ, 1.7^\circ, 2^\circ, 2.5^\circ, 3^\circ,$ and 3.5° , which correspond to the initial θ values in the CIMS process in Fig. 5 of the revised manuscript.

We have included these analyses in both the revised manuscript and supplementary information.

- ✧ Page 12, line 229: The sentences “We found that the H^{ext} with magnitude of 25 – 400 Oe tilted the average magnetization by approximately $0.5^\circ - 3.5^\circ$ from \mathbf{z} (Supplementary Information Section S6), but did not provide any preference for either the up or down magnetization state in the absence of the current injection” are added.
- ✧ Supplementary information: Section S6 is added for initial θ evaluation.

Figure R5. **a** Normalized AHE (i.e., $\cos \theta$) of a Pt/Co/SiO₂/Pt device measured at 300 K. The H^{ext} is applied at $\beta = 3^\circ$ and the injected current magnitude is 0.1 mA. **b** $\cos \theta$ vs. H^{ext} curve extracted from the first quadrant of **a**. **c** Variation of θ as a function of H^{ext} calculated from **b**.

5 While there is some deviation between the data and fit curve in Fig. 2c, they coincide almost completely in Fig. 2d. Why?

Response:

We think that this difference may be caused by the difference in the data scale. The data scale of $H_-^{\text{ext}} + H_+^{\text{ext}}$ (kOe) is remarkable larger than that of $H_-^{\text{ext}} + H_+^{\text{ext}}$ (Oe) (please refer to Figs. 3c, 3d in the revised manuscript). As a result, it seems that there is no deviation between the $H_-^{\text{ext}} + H_+^{\text{ext}}$ data and fit curve. However, a look to the magnification of the $H_-^{\text{ext}} + H_+^{\text{ext}}$ data marked with the dashed box in Fig. R5a clearly indicates that there is also a deviation between the $H_-^{\text{ext}} + H_+^{\text{ext}}$ data and fit curve (Fig. R5b).

Figure R5. **a**, $H_-^{\text{ext}} + H_+^{\text{ext}}$ as a function of $\cos \theta$. **b**, Enlarged image of the data points inside the dashed box in **a**.

6 At line 133, how is the formula of the efficiency obtained?

Response:

The formula used for antidamping torque efficiency (ζ^{AD}) estimation is obtained and modified from Khvalkovskiy *et al.* (*Phys. Rev. B* 87, 020402 (2013)) and Pai *et al.* (*Phys. Rev. B* 92, 064426 (2015)). In these studies, they ascribed the current-induced antidamping torque to the spin Hall effect while in our study, we considered the current-induced orbital polarization as the antidamping torque origin. Regardless of detailed mechanism, ζ^{AD} can be estimated by the formula $\zeta^{\text{AD}} = (2e/\hbar)M_{\text{stCo}}^{\text{eff}}(T^{\text{AD}}/j_{\text{PtCo}})$. We have added the two papers in the revised manuscript as references.

7 At line 147, it is written that T^{AD}_4 decreases with decreasing temperature but Fig.3c does not look so. Also, it is written that H^{an}_2 increases as temperature is decreased but Fig.3d does not look so.

Response:

The sentence “ T^{AD}_2 and T^{AD}_4 decrease with decreasing temperature, while H^{an}_2 and H^{an}_4 increase as temperature is decreased (H^{an}_2 shows an exception at around 125 K, where a peak is present)” at line 147 of the previous version of the manuscript is changed to “With regard to the high-order terms, T^{AD}_2 and the absolute value of T^{AD}_4 decrease with decreasing temperature; the absolute value of K_2 shows a not strictly monotonic increase with decreasing temperature and presents a salient at ~125 K, while K_3 have a monotonic temperature dependence” in the revised manuscript (Page 11, line 214).

8 At line 182, it would be helpful to add the definition of sp-metals.

Response:

We have improved the words “*sp*-metal” to “BiAg₂ monolayer with surface *s-p* orbital hybridization” in the revised manuscript (Page 15, line 306).

Reviewers' comments:

Reviewer #1 (Remarks to the Author):

I refer to my first review report. Based on the comments made there, in their answers and in the changed manuscript, the authors try to convince the reader, that the films show a homogeneous rotation of the magnetization upon application of the current in the Pt/Co bilayer.

This, however, cannot be true. As shown in Fig. 2, the coercive field H_c reaches an apparent value of zero at temperatures between 400C and 450C. This means that at this temperature the switching between up- and down-state has no energy barrier to overcome that is larger than the thermal energy.

In other words, one would expect a critical current density of zero for spin-orbit- or spin-transfer-torque switching and a correspondingly diverging efficiency for this temperature. This is, however, not due to a deep or new physical mechanism or effect – it is just thermally assisted switching at the edge to superparamagnetism.

This is further supported by the data, the authors give for the saturation magnetization in Fig. 2. There, M_s drops from its bulk-like value at low temperatures to less than half of this value at 400K. Bulk Co should have a T_c of around 1400K, and a 1nm thick homogeneous film should not deviate much from that. Thus at 400K, one would expect not more than about 15-20% reduction of M_s .

Thus in summary, I cannot find any evidence for the claim of the authors that the switching with SOT is by a coherent rotation of the magnetization, while it is domain wall nucleation and movement when applying a magnetic out-of-plane field. The fact, that the rotation is apparently coherent for an in-plane field does not mean that it will be also coherent when applying an SOT torque. This "in-plane-coherence" is just due to the fact that the perpendicular anisotropy is large enough to keep the magnetization out-of-plane either up or down. The switching between up- and down-state, however is still most probably due to the movement of domain walls with low energy barrier. Therefore, I still cannot recommend the publication in Nature Communications.

Reviewer #2 (Remarks to the Author):

Authors have addressed the points raised in the previous review with varying degrees of satisfaction. These results should stimulate the theoretical and experimental studies for interfacial spintronics. So I would be happy to see the manuscript accepted for publication at Nature Communications.

Reviewer #3 (Remarks to the Author):

The authors have revised the manuscript according to my comments. I recommend publication of this manuscript.

Responses to reviewers' comments

Manuscript ID: NCOMMS-17-25493B

Dear reviewers,

We appreciate your time and effort to read the manuscript and response letter. We have studied the comments carefully and replied them in the following with a point-by-point manner.

Reviewers' comments:

Reviewer #2 (Remarks to the Author):

Authors have addressed the points raised in the previous review with varying degrees of satisfaction. These results should stimulate the theoretical and experimental studies for interfacial spintronics. So I would be happy to see the manuscript accepted for publication at Nature Communications.

Reviewer #3 (Remarks to the Author):

The authors have revised the manuscript according to my comments. I recommend publication of this manuscript.

Response to reviewer #2 and #3: Your recommendation of acceptance and helpful comments in the reviewing process are highly appreciated!

Reviewer #1 (Remarks to the Author):

Comment 1: *I refer to my first review report. Based on the comments made there, in their answers and in the changed manuscript, the authors try to convince the reader, that the films show a homogeneous rotation of the magnetization upon application of the current in the Pt/Co bilayer. This, however, cannot be true.*

Response 1: Thanks for the reviewer's comments. We totally agree with the reviewer that the current-induced magnetization switching between up and down state is by the domain wall

motion, which is well recognized by abundant literature. In fact, we **have expressed our agreement and addressed this comment in the 1st response letter**. For example, in page 5 paragraph 2 in the 1st response letter, we have the sentences “In particular, we like to point out that the magnetization rotation mode is different in the measurements for CIMS and torque efficiency evaluation. In the CIMS measurement, the magnetization switching is achieved by the nucleation and propagation of domain walls rather than by coherent magnetization rotation”.

We would also like to point out that **we have never claimed coherent rotation of the magnetization to be the mechanism of the current-induced switching**. In page 8 lines 150–152 in the 1st revised manuscript (MS ID: NCOMMS-17-25493A), we have the sentences “This measurement scheme is introduced to prevent domain formation, such that the magnetization can rotate coherently and a macrospin approximation method can be applied to model the interplay between \mathbf{T}^{AD} and the torques exerted by \mathbf{H}^{ext} and the anisotropy field \mathbf{H}^{an} ”. There, we were saying that the magnetization rotates coherently in the torque efficiency measurement (Fig. 3a in the manuscript), rather than saying that the magnetization switching between up and down (Fig. 5a of the manuscript) is by the coherent rotation of the magnetization. We are afraid that it may have misled the reviewer. Accordingly, we modified the manuscript as below:

- ✧ In page 12 lines 236–238 in the 2nd revised manuscript (MS ID: NCOMMS-17-25493B), we added the sentence “Similar to the field-driven switching (Figs. 2a, 2b), the current-driven loops shown here also exhibit sharp-switching shape, which is due to the domain wall nucleation and propagation”.

Comment 2: *As shown in Fig. 2, the coercive field H_c reaches an apparent value of zero at temperatures between 400C and 450C. This means that at this temperature the switching between up- and down-state has no energy barrier to overcome that is larger than the thermal energy. In other words, one would expect a critical current density of zero for spin-orbit- or spin-transfer-torque switching and a correspondingly diverging efficiency for this*

temperature. This is, however, not due to a deep or new physical mechanism or effect – it is just thermally assisted switching at the edge to superparamagnetism. This is further supported by the data, the authors give for the saturation magnetization in Fig. 2. There, M_s drops from its bulk-like value at low temperatures to less than half of this value at 400K. Bulk Co should have a T_c of around 1400K, and a 1nm thick homogeneous film should not deviate much from that. Thus at 400K, one would expect not more than about 15-20% reduction of M_s .

Response 2: Thanks for the reviewer's comments. We have already addressed your concern about the current-induced thermal effect in pages 4–5 in the 1st response letter. Details about the assessment of the thermal effect have also been supplemented in the Section S5 of the 1st supplementary information. We have also added the discussions in page 12 lines 246–252 in the 1st revised manuscript (MS ID: NCOMMS-17-25493A).

To sum up, **we have provided data which demonstrated that the current-induced thermal effect is very small in our experiment.** A 4-mA current injection, corresponding to a current density 6×10^6 A/cm², just causes a temperature increase of ~6 K above room temperature (here 300.1 K) (see Fig. R1 in the 1st response letter). Note that the current magnitude is 0.9 mA in the torque efficiency measurement (Fig. 3a of the manuscript), and the 4-mA current is large enough to switch the magnetization direction in the CIMS measurement (Fig. 5 of the manuscript). We therefore think that the current-induced thermal effect cannot have significant influence on the critical current density and torque efficiency evaluations.

We would also want to point out that the experiments shown in Fig. 2 were performed in a physical property measurement system and the temperature elevation was done **by using a heater, not by using current-induced thermal effect.** As mentioned above, the thermal effect is very small in our experiment and it is impossible to heat the Pt/Co/SiO₂/Pt films to such high temperatures. Additionally, a zero value of H_c is a necessary but not sufficient condition for superparamagnetic behavior. Ferromagnetic films with multiple domains at

remanence state can also exhibit a zero value of H_c (Phys. Rev. Lett. 69, 3385 (1992); Phys. Rev. B 64, 054418 (2001)). The magnetic hysteresis and anomalous Hall loops shown in Figs. 2c and 2d demonstrates clearly that the Pt/Co/SiO₂/Pt films remain ferromagnetic even at temperature of 400 K.

We next response the comment “*not due to a deep or new physical mechanism or effect*”. In the manuscript, we reported a giant, anisotropic, and strongly temperature-dependent antidamping torque in perpendicularly-magnetized Pt/Co/SiO₂/Pt heterostructures. Apparently, **the results cannot be interpreted by existing spin Hall effect of Pt bulk and/or spin Rashba effect at the Pt/Co interface**, because the bulk spin Hall effect predicted the torque to be isotropic and temperature-independent and the simple spin Rashba model was shown to generate mostly a field-like torque instead of antidamping torque. It is well known that electrons have orbital in addition to spin, and the orbital moments are highly anisotropic and strongly temperature-dependent, hinting that the orbital of electrons may be the key to understand the results. Moreover, in 3d ferromagnet where the 3d orbitals couple with the lattice directly and the spin-orbit interaction is not strong, the crystal electric field should split the orbital primarily, and the spin-orbit interaction will act as a perturbation only (Fig. 6d of the manuscript). **We therefore proposed an orbital Rashba-Edelstein effect, the key idea of which is that the interfacial orbital hybridization can produce chiral orbital textures in momentum-space and cause an orbital polarization with current injection.** This effect is based on the intrinsic 3d band structures and does not require transport of the current-induced polarization. We found that the three main results of our work, i.e., the giant magnitude, the anisotropy, and the strong temperature dependence of the torque, can be well understood within the orbital Rashba-Edelstein effect, demonstrating its validity in explaining the in-plane current-induced torque in HM/FM/oxide heterostructures.

The significance of our finding is twofold.

Firstly, **we indicate a new route to engineer and optimize the in-plane current-induced torque in HM/FM/oxide heterostructures, that is, by manipulating the**

orbital of electron. For example, the 3d shell contains five orbitals: d_z^2 , d_{xz} , d_{yz} , d_{xy} , and $d_{x^2-y^2}$, and each 3d orbital could accommodate zero, one, or two electrons. The electron configurations are governed by Pauli exclusion principle and Hund's rules and are highly sensitive to the interface orbital hybridization in ultrathin ferromagnet. Therefore, compared to the spin of electron, the orbital of electron is more flexible for manipulation, for example, by engineering the ligand (here is oxygen atom) environment around the 3d atom using electric-field-driven oxygen migration effects (Mater. Sci. Eng., R 83, 1 (2014); Nature Mater. 14, 174 (2015)), which could lead to strategies for designing novel devices. In the manuscript, we reported a large torque efficiency of up to 2.83 and a low switching current density of order of 10^6 A/cm². As far as we know, this torque efficiency value is the largest one in reported HM/FM/oxide systems, and we believe that it can be further improved by carefully tuning the interface orbital hybridizations at both the HM/FM and FM/oxide interface.

The second significance of our findings is that our results may advances the current understanding of the in-plane current-induced torque in HM/FM/oxide heterostructures and has potential implications for theoretical calculations. Conventional wisdom about the effects of the interface orbital hybridization on ultrathin ferromagnets is that the interface orbital hybridization can enhance the orbital moments and cause the perpendicular magnetic anisotropy (Phys. Rev. Lett. 81, 5229 (1998); Phys. Rev. B 84, 054401 (2011)). Apparently, orbital hybridizations can modify band structures. However, whether and how the interface orbital hybridization affects the current-induced torque in ultrathin ferromagnets remains an open question. As mentioned above, compared to the existing spin Hall and spin Rashba models, the proposed orbital Rashba-Edelstein model has advantages in understanding the large, anisotropic, and strongly temperature-dependent torque we observed. Our study therefore presents improvements and/or alternatives to the existing models. We acknowledge that the proposed orbital Rashba-Edelstein model has weakness. For example, it fails to explain why the torque and magnetic anisotropy inversely scale with the polar angle (see Fig. 3e in the manuscript). Nevertheless, we believe that our results will stimulate theoretical studies where a realistic description of the electronic structures must be included.

To make the significance of our findings clearer, we have revised and added some sentences in the Conclusion section of the manuscript:

✧ Page 18, lines 273–278: “We have reported a large, anisotropic, and strongly temperature-dependent current-induced antidamping orbital torque in a Pt/Co/SiO₂/Pt heterostructure with high PMA. The results cannot be simply understood using the existing SHE and SRE but can be qualitatively interpreted within an OREE model we proposed, where the antidamping torque originates from intrinsic 3d band structures of the Co layer and does not require transport of the current-induced nonequilibrium polarizations”

Comment 3: *Thus in summary, I cannot find any evidence for the claim of the authors that the switching with SOT is by a coherent rotation of the magnetization, while it is domain wall nucleation and movement when applying a magnetic out-of-plane field. The fact, that the rotation is apparently coherent for an in-plane field does not mean that it will be also coherent when applying an SOT torque. This "in-plane-coherence" is just due to the fact that the perpendicular anisotropy is large enough to keep the magnetization out-of-plane either up or down. The switching between up- and down-state, however is still most probably due to the movement of domain walls with low energy barrier. Therefore, I still cannot recommend the publication in Nature Communications.*

Response 3: Please refer to response 1.

Thank you again for your time and effort. We hope that the responses have addressed your concerns clearly, and you find them satisfactory.

Reviewers' comments:

Reviewer #1 (Remarks to the Author):

The authors did reply to my comments in a way that could make the paper publishable. I am still not convinced that the evaluation of a "giant torque" is correct. The paper mixes macrospin models with a large and uniform effective perpendicular anisotropy with experimental results that are based on the nucleation and motion of domain walls. Thus for the magnetic switching process, one has to use in a very simple approach an effective anisotropy that is reduced corresponding to the values of the measured coercive field. A better approach is to evaluate the lifetime of a certain magnetic state under the presence of an effective driving force such as spin transfer torque or external field. Then, one has in a correct description a lifetime $\tau = \tau_0 * \exp((\Delta E - T * S - \mu * N) / k_B * T)$, where S is the entropy $S = k_B * \ln(W)$ and N is the number of particles (phonons and magnons in this case). W gives the probability of the state or to be more correct the number of possible pathways for magnetization switching. Then it turns out, that in particular the entropy dominates the lifetime for a magnetization switching by domain nucleation and growth and not the energy barrier ΔE , because there can be an enormous number of paths for this (see, e.g. Wild, J. et al., Science Advances 3, e1701704 (2017), where the number of pathways is of the order of $10^{20} - 10^{30}$). This will make all results dependent on the temperature and on the time that is needed for the measurement: fast cycling of the field will give larger coercive fields, short pulses will give a much larger critical current and so on.

Reviewer #3 (Remarks to the Author):

The authors have replied to the comments and revised the manuscript properly. I recommend publication of this manuscript.

Responses to reviewers' comments

Manuscript ID: NCOMMS-17-25493C

Dear reviewers,

We sincerely thank the reviewers for the constructive comments on the manuscript. Accordingly, the revised manuscript has been improved with additional interpretations. Modified sentences/words in the revised manuscript are highlighted in red. We respond below in detail to your comments. We hope you find the responses and revisions satisfactory.

Reviewers' comments:

Reviewer #1 (Remarks to the Author):

The authors did reply to my comments in a way that could make the paper publishable.

I am still not convinced that the evaluation of a "giant torque" is correct. The paper mixes macrospin models with a large and uniform effective perpendicular anisotropy with experimental results that are based on the nucleation and motion of domain walls. Thus for the magnetic switching process, one has to use in a very simple approach an effective anisotropy that is reduced corresponding to the values of the measured coercive field. A better approach is to evaluate the lifetime of a certain magnetic state under the presence of an effective driving force such as spin transfer torque or external field. Then, one has in a correct description a lifetime $\tau = \tau_0 \exp((\Delta E - T \cdot S - \mu \cdot N) / (k_B T))$, where S is the entropy $S = k_B \cdot \ln(W)$ and N is the number of particles (phonons and magnons in this case). W gives the probability of the state or to be more correct the number of possible pathways for magnetization switching. Then it turns out, that in particular the entropy dominates the lifetime for a magnetization switching by domain nucleation and growth and not the energy barrier ΔE , because there can be an enormous number of paths for this (see, e.g. Wild, J. et al., Science Advances 3, e1701704 (2017), where the number of pathways is of the order of $10^{20} - 10^{30}$). This will make all results dependent on the temperature and on the time that is needed for the measurement: fast cycling of the field will give larger coercive fields, short pulses will give a much larger critical current and so on.

Response:

We appreciate the positive feedback and the constructive comments.

At present, the macrospin model is the most straightforward and wide-spread method for evaluating the in-plane current-induced torque in heavy-metal (HM)/ferromagnet (FM)/oxide films with perpendicular magnetic anisotropy (see, e.g., Science 336, 555 (2012); Nat. Mater. 12, 240 (2013); Nat. Nanotech. 8, 587 (2013), and so on). The magnitudes of the torque efficiency (not the critical current density) from experiments based on the macrospin model have reasonable agreement with those from theoretical predictions. **We make the statement of “giant torque” based on the comparison of our torque efficiency with previously reported ones that were obtained also using macrospin models.** We would also like to say that the macrospin model presented in the manuscript is different from the typical one that is often used to simulate magnetization dynamics with assumption of a coherent magnetization switching. In our case, the external magnetic field is intentionally applied along the hard magnetization axis (not the easy magnetization axis) so that the magnetization can rotate coherently as a single spin without domain formation. That is, no magnetization switching occurs during the torque evaluation. We fear that we have misled the reviewer by the same name. **To avoid confusion with the typical macrospin model, we have rephrased the words “macrospin model” to be “macrospin method”** (see the revisions in red in page 8 of the manuscript).

The macrospin method only provide static information of the current-induced torque acting upon static magnetization. To analyze the effects of the torque on the magnetization switching quantitatively, it is more appropriate to probe the temporal information of the torque, as the magnetization switching is usually achieved through fast domain wall formation and motion. **We agree with the reviewer’s standpoint** that a better approach for torque measurement is to evaluate the lifetime of a certain magnetic state under the presence of an effective driving force. As far as we know, such information about the time evolution of the

in-plane torque during magnetization switching is still missing. The reviewer does indicate a new route for torque characterization and provide a perspective for fully understanding the origin of the in-plane torque in HM/FM/oxide heterostructures, which remains under debate.

We have added/revised some sentences in the manuscript to clarify this point and to recommend the new approach indicated by the reviewer for future studies:

- ✧ Page 12 lines 242–249: “We would like to emphasize that although the ζ^{AD} amplitude is often related to the j_c magnitude (Eq. 1), the large ζ^{AD} is not the exclusive causal factor for the low j_c values obtained here, because there can be an enormous number of pathways to overcome the energy barrier for magnetization switching³⁸, such as the domain wall formation and the time that is needed for the measurement. To analyze the effects of the current-induced torque on switching magnetization quantitatively, a better approach is to investigate the temporal information of the torque by measuring the lifetime of magnetization reversal under the presence of an electric current^{38–40}” are added.
- ✧ Page 11 lines 219–222: To examine the CIMS, we applied a small, fixed H^{ext} along \mathbf{y} or $-\mathbf{y}$, i.e., $\beta = 0^\circ$ or 180° , and then swept the quasistatic DC current with a ramp rate of 0.5 mA/s. The CIMS experiment was performed at 300 K and monitored by measuring R_{AHE} . (Modified from page 11 lines 223–225 in the previous manuscript “In order to confirm the large T^{AD} and ζ^{AD} , we performed CIMS experiments at 300 K, where the magnetization direction was monitored by measuring R_{AHE} using a quasistatic DC current sweeping method with a ramp rate of 0.5 mA/s”.)
- ✧ Abstract section: which is several times to one order of magnitude larger than those previously reported. (Modified from the Abstract section of the previous manuscript “thereby enabling a low j_c of the order of 10^6 A/cm²”.)

Moreover, we have gone through the manuscript carefully and revised some sentences/words to make the manuscript clearer and more coherent. Please refer to the accompanying table below.

Reviewer #3 (Remarks to the Author):

The authors have replied to the comments and revised the manuscript properly. I recommend publication of this manuscript.

Response: We appreciate your time and recommendation of acceptance! We have removed, added, and rephrased some sentences/words in the manuscript to avoid misunderstanding and to make the manuscript more coherent. Please refer to the responses to the reviewer #1 and the accompanying table below.

List of changes

Previous version (MS ID: NCOMMS-17-25493B)	Revised version (MS ID: NCOMMS-17-25493C)	Notes
	Fig. 6 reconfigured	To make the manuscript more readable as well as to meet the format requirements
	Refs. 38, 39, 40 added	
Abstract: The concurrence of high perpendicular magnetic anisotropy (PMA) and low critical current density (j_c) for current-induced magnetization switching in ferromagnetic heterostructures is of great importance for the realization of high-density and low-power-consumption magnetic devices. To achieve this goal, obtaining large current-induced torque efficiency is critical. However, the torque efficiency reported thus far has been usually low, rarely exceeding 0.3.	Shortened to “Enhancing the in-plane current-induced torque efficiency in inversion-symmetry-breaking ferromagnetic heterostructures is of both fundamental and practical interests for emerging magnetic memory device applications”	To meet the format requirements for manuscript
Abstract: advances the development of high-performance magnetic devices	Rephrased to be “indicates a new route for torque efficiency optimization through orbital engineering”	To underline the significance of the work

Abstract: a mechanism	Rephrased to be “interface-originated magnetoelectric effect”	
	Page 4 lines 69–70: “The torque is anisotropic with respect to the magnetization direction and shows strong temperature dependence” added.	
Page 4 lines 71, 72: which may improve the understanding of the in-plane current-induced torque in HM/FM/oxide heterostructures	Deleted	Repeated words
Page 5 lines 87–89: We also notice that the Co disperses in a range that is much wider than its thickness, which we attribute to the drift during the mapping image acquisition and atom diffusion	Deleted	Unnecessary words
Page 6 lines 99–105	Moved to page 6 lines 102–105 and revised	To make the manuscript more readable and understandable
Page 6 lines 111–126	Moved to page 6 115–125 and revised	
Page 9 line 166: the validity of which has been verified by a number of studies	Deleted	Unnecessary words
	Page 8 line 142: i.e., almost along the hard magnetization axis	Added to make the point clearer
Page 12 lines 248–256	Moved to page 10 lines 199–205 and revised	To make the manuscript more

		readable and understandable
	Subheadings of the Discussion section deleted	To meet the format requirements for manuscript
Page 13 line 262: The size of ζ^{AD} is proportional to the θ_{SH} within the SHE model	Deleted	Repeated words
Page 14 lines 278–288	Revised	To make the point we want to express clearer
	Page 15 lines 296–298: As already mentioned, the spin states are dominated by the orbital splitting and, thus, in turn, can be taken as a gauge of the size (Δ_{os}) of the orbital splitting	Added to make the point clearer
Page 15 line 297: followed by a further split into states of spin via the SOC	Page 14 lines 282–283: while the SOC, which is not strong in Co metal, only acts as a perturbation to the band energy governed by the orbital splitting and further splits them into states of spin	Revised to make the sentences understandable
	Page 15 lines 309–310: As the Δ_{os} also dominates the amplitude of the current-induced $\Delta\mathbf{L}$	Added to make the point clearer
Page 15 lines 309–312: Different from the nonmagnetic systems, the	Page 14 lines 290–292: Contrast to the nonmagnetic tellurium	Revised to make the point

generated $\Delta\mathbf{L}$ in the magnetized Pt/Co/SiO₂/Pt films can further couple to the Co magnetization via exchange interaction, thereby producing a torque upon the magnetization and eventually inducing magnetization switching at a critical current	crystals with bulk inversion asymmetry, the current-induced $\Delta\mathbf{L}$ in our films is interface-originated and can further couple to and exert torques upon the Co magnetization via exchange interaction.	understandable
Page 16 lines 337–339: In nonmagnetic systems, Δ_{os} is unalterable because band structures are fixed in the absence of additional external forces, such as temperature variation	Deleted	Insignificant words
Page 17 lines 339–341: In magnetized systems, by contrast, band structures change with magnetization rotation due to the intrinsic OM anisotropy and the Fermi surface distortion induced by the additional exchange field	Page 15 lines 313–314: In magnetized systems, the band structures depend not only on the intrinsic Δ_{os} but also on the magnetization direction, due to exchange field-induced Fermi surface distortion effect	Rephrased to make the point more understandable
Page 17 lines 341–342: The former produces an angular-dependent OM magnitude and the latter can modify the energy scale	Deleted	Repeated words
Page 18 lines 366–368: We notice that the correction between the high-order terms in T^{AD} and K_u is only semiquantitative. For example, in	Deleted	Repeated words

Fig. 4b, K_2 presents a salient at ~ 125 K, but in Fig. 4a, this feature is absent in T_2^{AD} and T_4^{AD} .		
Page 18 lines 380–381: suitable for magnetization switching with low j_c	Deleted	To avoid misunderstanding
Page 19 lines 384–385: whose origin cannot be explained solely by the SHE and SRE	Deleted	Repeated words
Page 19 lines 386–388: the multiplicity of the orbital symmetries and their sensitivity to both crystal fields and interfacial orbital hybridization provide great opportunities to realize a nonvolatile voltage-programmable orbital torque by means of electric-field-driven ionic migration effects	Page 17 lines 353–355: Rephrased to be “the multiplicity of the orbital symmetries and their sensitivity to the interfacial orbital hybridization provide great opportunities to optimize the torque efficiency and critical switching current, for example, by orbital engineering using electric-field-driven ionic migration effects”	

Reviewers' Comments:

Reviewer #1 (Remarks to the Author):

I still think that the results are not evaluated in a correct way. The fact that in most of the literature the same incorrectness is used is at my point of view not a good argument to follow the same misleading path. In particular, there is no "giant" torque, because only a small coercive field has to be overcome by a correspondingly small torque.

The changes, the authors made do now give some (weak) hints that there could be a better way to correctly evaluate the data. This makes the paper now publishable.

Responses to reviewers' comments

Manuscript ID: NCOMMS-17-25493D

Dear reviewer #1,

We sincerely thank you for your time and comments. Below are the detailed responses to your concerns.

Reviewers' comments:

Reviewer #1 (Remarks to the Author):

I still think that the results are not evaluated in a correct way. The fact that in most of the literature the same incorrectness is used is at my point of view not a good argument to follow the same misleading path. In particular, there is no "giant" torque, because only a small coercive field has to be overcome by a correspondingly small torque.

The changes, the authors made do now give some (weak) hints that there could be a better way to correctly evaluate the data. This makes the paper now publishable.

Response:

Thanks for the comments. We admit that while the static macrospin method is straightforward and widespread, it cannot reflect the dynamic information of the current-induced torque. Discrepancy between static and dynamic torques may be expected, which is an interesting topic for future study. In the last two weeks, we got some data about orbital engineering of the torques in Pt/Co/oxide heterostructures in our recently undergoing works. We found that the electron states of the Co 3d orbitals near the Co/oxide interface can be tuned when different oxide layer was used. More interestingly, we found corresponding changes in the current-induced torque, including its magnitude and symmetry; the formation of Co²⁺ favors an enhanced torque while that of Co³⁺ does the opposite. These new results provide further supports of our statements that the orbital hybridization at the ferromagnet/oxide interface play a critical role in obtaining the large torque in our Pt/Co/SiO₂/Pt films (the Co near the

Co/SiO₂ interface has a 2+ valence). Additionally, we found no direct correlation between the coercivity field and the torque in different Pt/Co/oxide films. We hope that our works could help to understand and manipulate the torques in the view point of orbital-driven variation of electronic structures.

We have revised some words/sentences in the manuscript to make the paper clearer:

- ✧ Page 3 lines 50,51: “when the device size is varied down to the scale that does not accommodate domain wall formation” is added.
- ✧ Paragraph 3 in page 12 is revised based on the above responses (please see the text in red in the revised manuscript).

Thank you again for your attention and the recommendation of publication.